# Mesenchymal glioma stem cells trigger vasectasia–distinct neovascularization process stimulated by extracellular vesicles carrying EGFR

Cristiana Spinelli[1], Lata Adnani[1], Brian Meehan[1], Laura Montermini[1], Sidong Huang[2,3], Minjun Kim[4], Tamiko Nishimura[4], Sidney E. Croul[5], Ichiro Nakano[6], Yasser Riazalhosseini[4] & Janusz Rak[1,2,4] ✉

Targeting neovascularization in glioblastoma (GBM) is hampered by poor understanding of the underlying mechanisms and unclear linkages to tumour molecular landscapes. Here we report that different molecular subtypes of human glioma stem cells (GSC) trigger distinct endothelial responses involving either angiogenic or circumferential vascular growth (vasectasia). The latter process is selectively triggered by mesenchymal (but not proneural) GSCs and is mediated by a subset of extracellular vesicles (EVs) able to transfer EGFR/EGFRvIII transcript to endothelial cells. Inhibition of the expression and phosphorylation of EGFR in endothelial cells, either pharmacologically (Dacomitinib) or genetically (gene editing), abolishes their EV responses in vitro and disrupts vasectasia in vivo. Therapeutic inhibition of EGFR markedly extends anticancer effects of VEGF blockade in mice, coupled with abrogation of vasectasia and prolonged survival. Thus, vasectasia driven by intercellular transfer of oncogenic EGFR may represent a new therapeutic target in a subset of GBMs.

Glioblastoma multiforme (GBM) is the most frequent and deadly form of primary astrocytic brain tumour[1] associated with exuberant and heterogeneous vascularity[2]. The GBM 'growth units' consist of glioma stem cells (GSCs)[3], the transcriptomes of which range from proneural (PN) to mesenchymal (MES) profiles[4,5], reminiscent of the corresponding GBM subtypes[1,6]. These characteristics result from the superimposition of differentiation programmes[7] and underlying effects of oncogenic mutations[8]. Paradigmatic in this regard, is the amplification of the epidermal growth factor receptor (EGFR) along with its activating truncation yielding a variant (EGFRvIII) impacting both, intrinsic and non-cell-autonomous aspects of tumourigenicity[9].

Notably, EGFR/EGFRvIII is often expressed in MES-GSCs, but not in their PN-GSC counterparts[4,10], a dichotomy with poorly understood biological consequences.

Tumour initiation by GSCs occurs within a complex tissue microenvironment[11] including perivascular and parenchymal niches[12–14]. Conversely, GSCs are thought to exhibit an enhanced proangiogenic activity, including the elevated production of the vascular endothelial growth factor (VEGF), amidst other influences responsible for pathological vascular growth and dysmorphia that accompany GBM progression[2,15]. While therapeutic targeting of the VEGF angiogenesis pathway produced marked vascular responses in

[1]McGill University, The Research Institute of the McGill University Health Centre, Montreal, QC, Canada. [2]Department of Biochemistry, McGill University, Montreal, QC, Canada. [3]Rosalind & Morris Goodman Cancer Institute, McGill University, Montreal, QC, Canada. [4]Department of Human Genetics, McGill University, Montreal, QC, Canada. [5]Department of Pathology & Laboratory Medicine, Dalhousie University, Halifax, NS, Canada. [6]Department of Neurosurgery, Hokuto Social Medical Corporation, Hokuto Hospital, Kisen-7-5 Inadacho, Obihiro, Hokkaido 080-0833, Japan. ✉e-mail: janusz.rak@mcgill.ca

GBM patients[16], it did not improve survival[1]. These surprising outcomes point to remaining gaps in the current understanding of the GBM neovascularization mechanisms[17] and their mediators[18].

Tumour neovascularization is strongly influenced by oncogenic pathways through their impact on the expression of both, soluble[19] and particulate vascular effectors[20]. Among the latter, a unique role is attributed to extracellular vesicles (EVs), membrane-bound cellular fragments released from cells as either exosomes, membrane-derived microvesicles/ectosomes, or apoptotic bodies, all capable of intercellular transmission of macromolecular cargo (proteins, RNA)[21], including active oncogenes[22]. Notably, EVs have been implicated in various aspects of GBM-related angiogenesis[20,23], including transport of VEGF[24], non-coding RNA[25] and other mediators[11], but the nature and contributions of these processes to GSC-vascular communication[26] remain poorly understood. Here we report that EGFR-carrying, MES GSC-derived EVs trigger VEGF-independent vascular responses manifested by dysmorphic, circumferential vascular growth, we refer to as *vasectasia*.

## Results

### Glioma stem cell subsets trigger distinct vascular patterns in vivo

Since PN and MES glioma stem cell subtypes markedly differ with regard to profiles of their associated angiogenic mediators[27] and EVs[26], we chose to examine these differences in greater functional detail. As reported earlier[4], we observed that PN-GSC-initiated intracranial xenografts in immune-deficient (NSG) mice progress at a slower rate than their MES-GSC-initiated counterparts (Fig. 1a). Remarkably, this disparity was also associated with stark differences in vascular patterning (Fig. 1b). Thus, while PN tumours exhibited dense networks of capillary sized (angiogenic-type) blood vessels positive for CD31, their more aggressive MES counterparts contained unexpectedly less dense vascular patterns dominated by larger vessels, up to 100–300 μm in diameter (Fig. 1c, d). As these structural features are inconsistent with characteristics of angiogenic capillary sprouting[28], we refer to this novel large blood vessel formation process as *vasectasia*.

### Extracellular vesicles from mesenchymal glioma stem cells transfer EGFR transcripts to endothelial cells

To glean more insights as to the nature of differential vascular effects evoked by PN-GSCs and MES-GSCs cells, their conditioned media (CM) were tested for the ability to induce endothelial cell responses. Thus, we measured endothelial cell outgrowths from mouse aortic ring explants (Fig. 1e–g; Supplementary Fig. 1a, b) along with migration of cultured human primary endothelial cells (HUVEC), and human brain microvascular endothelial cells (HBMVEC; Supplementary Fig. 1c–f) in transwell assays. These responses to tumour CM were measured against negative controls and high concentrations of recombinant VEGF (Fig. 1e–g; Supplementary Fig. 1c, d). All GSC lines exhibited endothelial stimulating activity in these assays albeit to somewhat variable degrees. In view of earlier reports suggesting angiogenic roles of both soluble (VEGF)[15] and insoluble (EV)[24] mediators, we subjected GSC-derived CM to differential centrifugation[26] and compared the activities of supernatants and small EV fractions (Fig. 1h–j; Supplementary Fig. 1d–f). Interestingly, in the case of PN-GSCs, endothelial-stimulating activity was largely (though not exclusively) contained in the culture supernatant, whereas MES-GSC released a considerable proportion of endothelial stimulators in association with EVs. As reported[15], the culture supernatants of all GSCs contained appreciable amounts of soluble VEGF, which was virtually undetectable in the corresponding EV fractions, including highly active MES-GSC EVs (Fig. 1k).

In the virtual absence of VEGF in MES-GSC EVs, we sought alternative mediators of endothelial stimulating activity. Since vascular responses to external stimuli often involve activation of receptor

tyrosine kinases (RTKs), we interrogated their phosphorylation status in human and mouse endothelial cells treated with GSC-EVs (Fig. 2a). Remarkably, PN-GSC EVs elicited relatively weak phosphorylation across the panel of 49 RTKs included in the immobilised antibody array (e.g., traces of pVEGFR1). On the other hand, MES-GSC-derived EVs triggered several RTKs, of which phosphorylated EGFR (pEGFR) overshadowed other responses (pVEGFR1, pALK, pVEGFR2, pAXL) by orders of magnitude. Notably, endothelial cell expression of pEGFR was selectively stimulated by MES-GSC EVs, but not by PN-GSC EVs (Supplementary Fig. 2a), as early as within 24 h of incubation and was sustained for up to 6 days post-treatment (Fig. 2b; Supplementary Fig. 2b). This is of interest as cancer cell EVs are known to carry oncogenic and bioactive EGFR protein[20,23,29], which can be internalised by endothelial cells leading to their reprograming[22]. Indeed, we observed the acquisition of EGFR expression in human large vessel-derived (HUVEC)- and brain microvessel-derived endothelial cells (HBMECs) in the presence of MES-GSC-EVs, but not in the case of PN-GSC-EVs (Supplementary Fig. 2b–d). These EVs were bioactive, however, as mouse endothelial cells (EOMA) exposed to PN-GSC-EVs acquired the expression of CD133 (Supplementary Fig. 2d), a known marker of PN-GSC[10].

It is possible that rather than transferring EGFR, the cargo of MES-GSC EVs may trigger the expression of endogenous EGFR by recipient endothelium[30]. To distinguish between these possibilities, we incubated human MES-GSC EVs with mouse endothelial cells (EOMA) and probed them specifically for human EGFR (hEGFR). Both hEGFR protein and transcript were readily detectable in GSC EVs and in EV-treated EOMA cells (Fig. 2c, d) suggesting intercellular transmission of this receptor. Moreover, when human endothelial cells were incubated with MES-GSC EVs, they became positive for GBM-specific mutant EGFRvIII mRNA and protein expression (Supplementary Fig. 2e) further enforcing the notion that endothelial cells express oncogenic EGFR received from cancer cells[22].

While these observations are intriguing, they do not specify whether EGFR protein or mRNA is responsible for the lasting expression of EGFR/pEGFR in EV recipient cells. We reasoned that the transfer of EGFR protein alone[22] (with reported EGFR half-life under 30 h[31]) would unlikely account for the 6-day long ectopic expression of this oncogenic receptor in endothelial cells. This may suggest a possible contribution of EGFR mRNA transfer[23] by a subset of tumour EVs. In this regard, a considerable heterogeneity of GSC EVs was previously extensively characterised highlighting the preponderance of neutral sphingomyelinases (SMPD2/3/4) in MES-GSCs and of exosomal markers in their related EVs[26]. To test the role of this EV biogenesis pathway[32], MSC-GSCs were treated with the neutral sphigomyelinase inhibitor (GW4869) and tested for the EV output and EGFR content. While the overall EV release was not changed post GW4869 treatment (Supplementary Fig. 3a) and their content of EGFR mRNA remained comparable, the corresponding levels of EGFR protein declined dramatically (Fig. 2e). These observations suggested the existence of several pathways (GW4869-sensitive and -insensitive) in MES-GSCs, which separately traffic EGFR protein or RNA into EVs and the pericellular milieu. To further explore this possibility, MES-GSC EVs were incubated with magnetic beads coated with anti-EGFR antibody to separate/deplete EVs expressing EGFR protein on their surfaces (Fig. 2f). This treatment efficiently removed the EGFR-antigen-positive EVs from the particulate secretome, while the remaining EVs (flow through) were enriched for EGFR mRNA. Remarkably, EVs containing EGFR and EGFRvIII mRNA (but depleted for EGFR/EGFRvIII proteins) efficiently transferred GSC-related EGFR and EGFRvIII to recipient endothelial cells and their effect persisted for several days (Fig. 2g). EGFRvIII mRNA was also directly detected in endothelial cells exposed to unfractionated MES-GSC-EVs (Supplementary Fig. 3b). These observations suggests that the EV-mediated intercellular transfer of

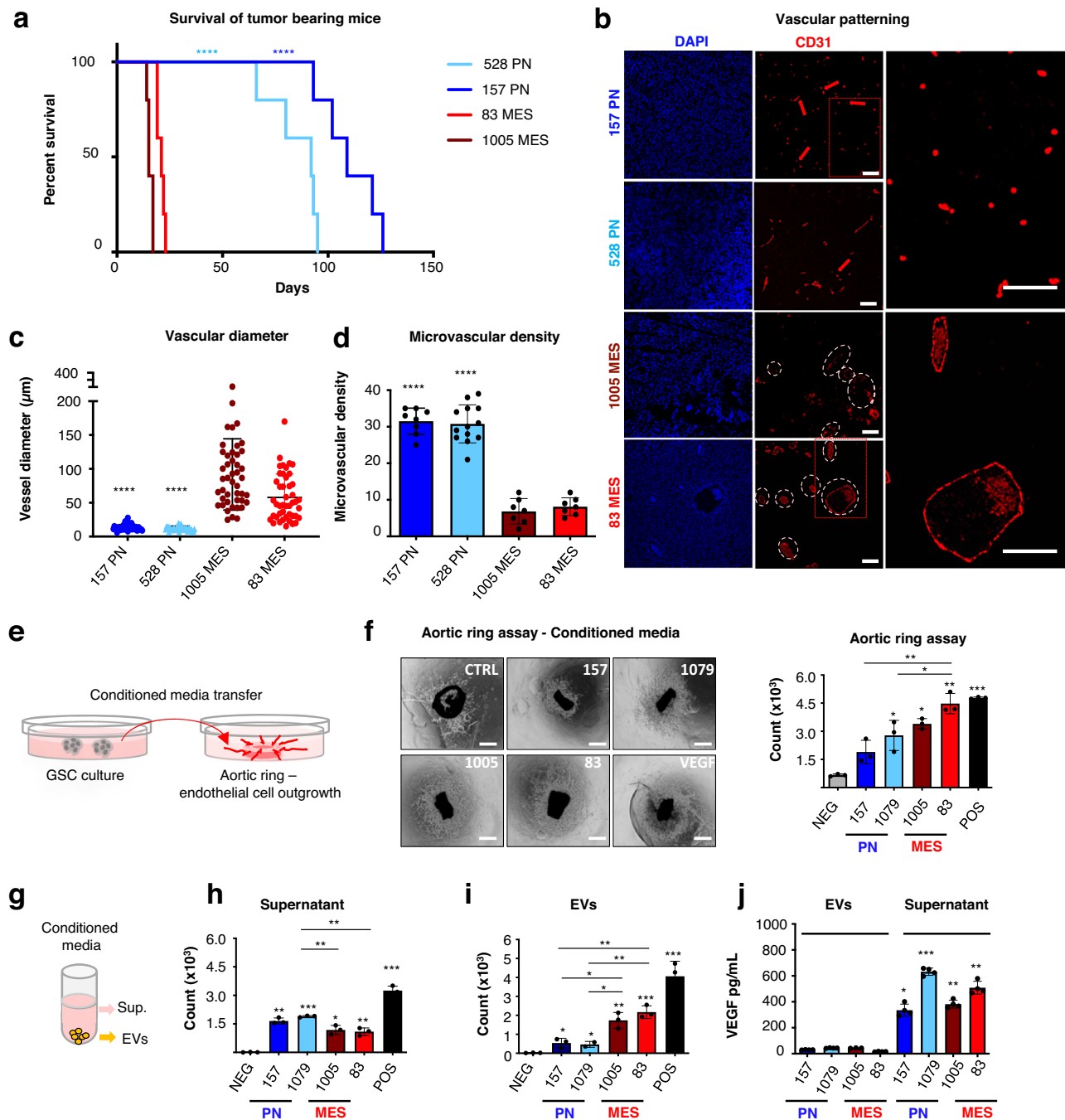

**Fig. 1 | Differential vascular patterns in GSC-driven tumours and vascular activities of soluble and vesicular components of glioma stem cell secretome.** **a** Kaplan-Meier survival curves of mice bearing PN GSC- and MES GSC-derived tumours. ($n = 5$ independent experiments. Two-tailed paired $t$ test. $P = 0.0000965$ and $0.000089$) **b** Representative images of immunofluorescence for CD31 reveal phenotypic vascular differences between tumours driven by PN or MES GSCs. ($n = 5$ independent experiments). **c** Quantification of vessel size distribution through tracing CD31 positive endothelial cells. Blood vessels in MES tumours present enlarged lumens, up to 90 μm, compared to a mean vessel diameter of 13 μm in the PN tumours ($n = 5$ mice/group. Two-tailed paired $t$ test $P = 5.48^{-16}$ and $1.18^{-12}$). **d** Quantification of microvascular density using CD31 staining ($n = 5$/group. Two-tailed paired $t$ test. $P = 5.25^{-09}$ and $2.68^{-09}$). Microvascular density was expressed as vessel density per high power field (hpf). Scale bars are 50 μm **e** Schematic diagram illustrating mouse aortic ring endothelial outgrowth assay using conditioned media derived from different glioma stem cells. **f** Endothelial responses induced by the GSC conditioned media containing soluble fraction of the secretome and EVs. RhVEGF was used as positive control [25 ng/mL]. Cells were imaged with optical microscope (left) to assess the number of endothelial cells growing out of the ring

using FIJI software (right) ($n = 6$ independent experiments. Two-tailed paired $t$ test. MES $P = 7.55^{-05}$ and $2.65^{-04}$). **g** Schematic diagram illustrating secretome fractions preparation using centrifugation methods. **h** Endothelial responses induced by the GSC secretome. Mouse aortic rings were seeded under domes of BME and cultured in growth factors-enriched media. After rings began to form endothelial outgrowth (often referred to as 'sprouts') they were treated with supernatant fractions (PN $P = 0.00037$ and $3.31609^{-06}$. MES $P = 0.0010$ and $0.00073$) or with (**i**) 30 μg/mL of EVs obtained from either PN GSC (157;1079), or MES GSC (83; 1005). RhVEGF was used as positive control [25 ng/mL]. The number of endothelial cell outgrowths from the ring was assessed using FIJI software. ($n = 6$ wells/3 independent experiments. Two-tailed paired $t$ test. PN $P = 0.0229$ and $0.0023$. MES $P = 0.0021$ and $0.00037$). **j** VEGF content distribution in EVs and in supernatant fractions of GSC conditioned media. ELISA assay quantification showed that the growth factor is virtually absent in EVs, and preferentially released in soluble form into the culture media supernatant ($n = 4$ independent experiments. Two-tailed paired $t$ test. PN $P = 1.21^{-05}$ and $1.30^{-08}$. MES $P = 8.08^{-06}$ and $9.83^{-07}$). Results are shown as mean ± SD; $*p < 0.05$, $**p < 0.01$, $***p < 0.001$ treated group versus control group $****P < 0.0001$; Detailed data are provided as a Source Data file.

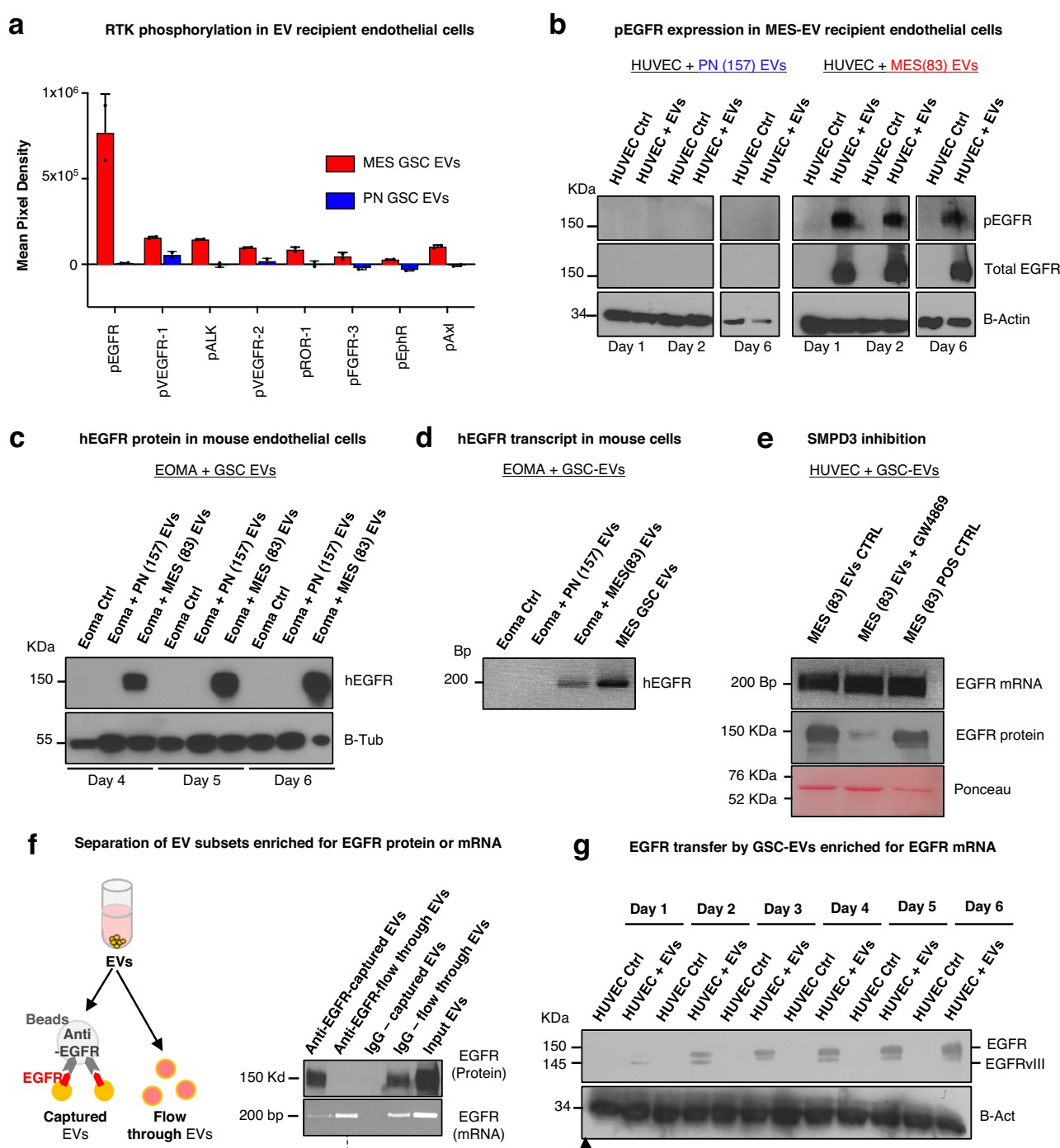

**Fig. 2 | Protracted expression of activated EGFR in endothelial cells subjected to extracellular vesicle-mediated transfer of EGFR/EGFRvIII transcript from mesenchymal glioma stem cells. a** Quantification of the human phospho-RTK expression array; analysis performed with EVs from either PN or MES EVs incubated with primary endothelial cells (HUVEC) for 7 days (*n* = 4 wells/2 independent experiments). Data are presented as mean values ± SD; **b** Endothelial cells incubated with EVs from PN or MES EVs followed by protein extraction and western blot to analyse the expression of EGFR. The transfer resulted in lasting ectopic activation of EGFR in HUVEC up to 6 days. β-Actin was used as loading control. (*n* = 3 independent experiments); **c** Glioma stem cell-derived EGFR+ EVs were incubated with mouse endothelial cells. The transfer of human-specific EGFRvIII was detected for up to 6 days. Mouse β-Tubulin was used as loading control (*n* = 3 independent experiments); **d** Glioma stem cell-derived EVs were incubated with mouse endothelial cells enabling transfer of human EGFR. The transfer of human-specific EGFR

mRNA was detected only after treatment with MES GSC EVs (*n* = 3 independent experiments); **e** Western blot and RT-PCR analysis showed that GW4869 treatment selectively inhibited the shedding of EVs carrying EGFR protein, while EGFR mRNA EVs were not affected (*n* = 3 independent experiments); **f** Schematic diagram illustrating the immunoprecipitation approach using magnetic beads crosslinked with anti-EGFR antibody. Western blot and RT-PCR analysis showed a population of EVs enriched for EGFR protein and mostly depleted for EGFR mRNA, while unbound EVs with no EGFR protein were enriched for EGFR mRNA (*n* = 3 independent experiments); **g** EGFR status in endothelial cells treated with EVs carrying only EGFR mRNA. Western blot analysis reveals that transfer of EGFR mRNA is sufficient to express EGFR protein in primary endothelial cells for up to 6 days (*n* = 3 independent experiments); Source data are provided as a Source Data file.

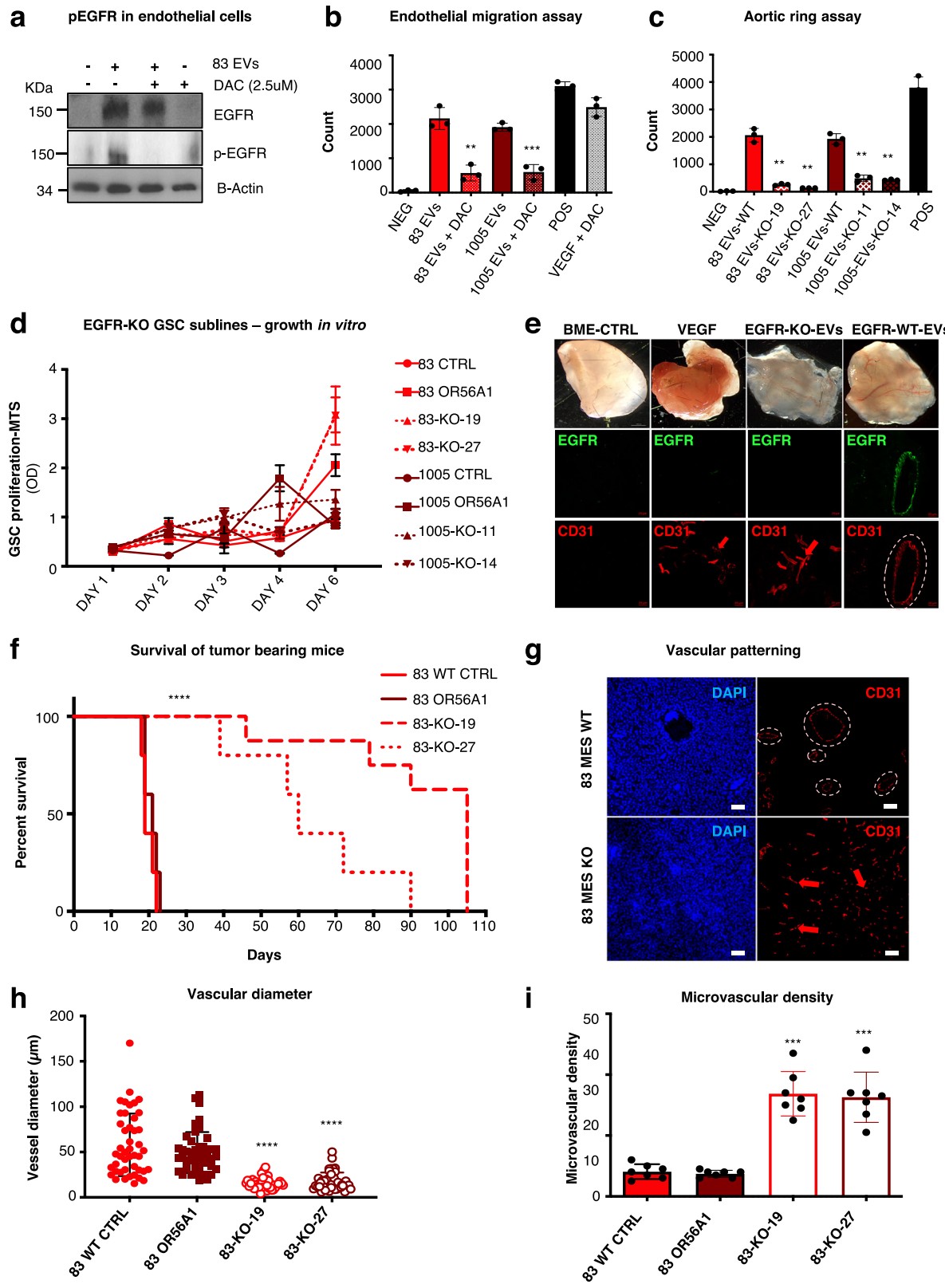

EGFR/EGFRvIII mRNA is sufficient to trigger the EGFR/EGFRvIII protein expression in endothelial cells.

## Intercellular transfer of EGFR activity leads to endothelial cell responses that parallel vasectasia

It could be expected that disabling the ectopic (EV-related) EGFR activity either before or after internalisation by endothelial cells, would block their responses to MES-GSC EVs. To assess this possibility, cultured endothelial cells were exposed to MES-GSC EVs and subsequently treated with Dacomitinib (DAC), a potent, clinically relevant, and irreversible panErbB (EGFR) inhibitor. Indeed, treatment with optimised concentrations of DAC (Supplementary Fig. 4a) reversed EGFR phosphorylation in endothelial EV recipient cells (Fig. 3a) and dramatically suppressed the MES-GSC EV-stimulated endothelial cell

**Fig. 3 | Obliteration of GSC-EV-derived EGFR in endothelial cells suppresses cellular responses to extracellular vesicles and alters vascular patterning in vivo. a** Activation of EGFR in endothelial cells following EV transfer from cancer cells is obliterated by the pan-ErbB inhibitor, Dacomitinib. EGFR phosphorylation in primary endothelial cells is completely inhibited by 2.5 μM of Dacomitinib treatment (*n* = 3 independent experiments); **b** Dacomitinib inhibition of endothelial cell migration triggered by EGFR-carrying EVs. Endothelial cells were treated with MES EVs or VEGF and 6 h later exposed to 2.5 μM of Dacomitinib. The number of cells migrated was assessed using FIJI software (*n* = 6 wells/3 independent experiments; two-tailed paired *t* test *P* = 0.0022 and 0.00077); **c** EGFR depletion reduces the ability of GSC EVs to trigger endothelial cell outgrowths. Endothelial cells were treated with 30 μg/ml of EVs obtained from glioma stem cells either deficient (EGFR-KO, clone 19 and 27), or proficient (EGFR-WT) for EGFR. After 3 days of incubation, cells were fixed, stained with crystal violet and imaged (*n* = 3 independent experiments; two-tailed paired *t* test; GSC83 (83): *P* = 0,00024 and 0,00018; GSC1005 (1005): *P* = 0.00041 and 0.00019); **d** Proliferation assay reveals similar growth pattern in culture of glioma stem cells deficient (EGFR-KO) or proficient (EGFR-WT) for EGFR (*n* = 3 independent experiments); **e** Appearance of

freshly removed BME plugs containing indicated agents three weeks after implantation. BME-embedded EVs were obtained from glioma stem cells: EGFR-KO, EGFR-WT, while control plugs contained VEGF, or vehicle. Scale bars are 20 μm (*n* = 4 independent experiments); **f** Kaplan-Meier survival curves of mice bearing EGFR-KO and EGFR-WT MES GSC-driven tumours (*n* = 5 mice per group); **g** Representative images of immunofluorescence for CD31 reveals differential vascular patterns between tumours driven by EGFR-WT or EGFR-KO MES-GSCs (GSC83). Scale bars are 20 μm; *n* = 5 independent experiments have been conducted; two-tailed paired *t* test *P* = 0,0008; **h** Quantification of vessel size distribution according to staining for CD31-positive endothelial cells (*n* = 5 independent experiments; two-tailed paired *t* test *P* = 1.45$^{-12}$ and 3.41$^{-11}$); **i** Quantification of microvascular density using CD31 staining (*n* = 5 independent experiments; two-tailed paired *t* test *P* = 1.39$^{-06}$ and 7.15$^{-06}$). Microvascular density was expressed as vessel density per high power field (hpf). Data were presented as means ± SD. Significance: **$p < 0.01$, ***$p < 0.001$, ****$p < 0.0001$ of the treated group versus untreated control group; Source data are provided as a Source Data file.

migration and formation of aortic ring outgrowths (Fig. 3b; Supplementary Fig. 4b). In line with these findings, depletion of EGFR in MES-GSC by CRISPR/Cas9-mediated editing of the *EGFR/EGFRvIII* gene (KO), resulted in a marked diminution of the EV-related ability to stimulate endothelial aortic ring outgrowths and endothelial cell migration (Fig. 3c; Supplementary Fig. 5a–d). Unexpectedly, *EGFR*-KO did not impact the proliferation of MES-GSC cells in complete growth media in vitro (Fig. 3d) suggesting that oncogenic EGFR is not essential for intrinsic control of growth and survival of these cells, but instead may possess non-cell-autonomous functions in GBM, at least in part, mediated by its EV-mediated export.

To explore these findings further, EVs from EGFR-proficient (WT) MES-GSCs were embedded in subcutaneous implants of the basement membrane extract (BME) in mice in the absence of cancer cells. In this setting, EGFR-carrying EVs elicited formation of vascular networks reminiscent of those observed in corresponding xenografts (Fig. 1b) and comprising unusually large vessels along with smaller endothelial structures (Fig. 3e; Supplementary Fig. 6). In contrast, EVs from the isogenic *EGFR*-KO MES-GSCs triggered mostly capillary ingrowths (Fig. 3e; Supplementary Fig. 6), similar to those observed in VEGF-containing BME pellets. These observations suggest that EGFR-EVs may be sufficient to trigger the peculiar vascular patterning associated with MES-GSC-driven gliomas.

Moreover, *EGFR* gene disruption dramatically curtailed the aggressiveness of MES-GSC intracranial xenografts in vivo (Fig. 3f; Supplementary Fig. 7a). In the absence of EGFR, these tumours also failed to develop large vessel patterns and instead contained dense capillary networks (Fig. 3g–i; Supplementary Fig. 7b–d), reminiscent of those observed in PN-GSC-initiated lesions (Fig. 1b). Finally, selective restoration of the oncogenic *EGFRvIII* in MES-GSC-KO cell lines (KO-EGFR+) resulted in a partial restoration of their tumourigenic phenotype, with increased aggressiveness, and reduced microvascular density coupled with enlarged vessel diameters (Supplementary Fig. 8a–d).

Collectively, these observations point to at least two different blood vessel growth processes induced by GSCs in vivo. In the absence of oncogenic EGFR (PN-GSCs, EGFR-KO-GSCs), GSCs produce mostly soluble angiogenic factors (including VEGF), which appear to drive the formation of dense capillary networks, ostensibly through a process of angiogenesis[2]. In contrast, MES-GSCs deploy EGFR/EGFRvIII-containing EVs that possess endothelial stimulating activity and trigger responses that may lead to circumferential extension of tumour blood vessels, a process that we termed *vasectasia*. In the latter case, our morphometric analysis of evolving vascular patterns post-tumour cell inoculation indicated a progressive increase in vessel diameter beginning at the boundary between brain parenchyma and MES-GSC-

driven tumour masses, a change that extended toward tumour interior. This response included the enlargement/remodelling of the same vessels at the point of their tumour entry and discernible as early as 5 days post cancer cell inoculation and evident by day 9 (Fig. 4a). Thus, the capillary brain vasculature external to the tumour contained mostly vessels of ~5 μm in diameter. Upon entry into the tumour, the same vessels expanded to reach diameters of 15 μm, or more, and further expanded to 25 μm and beyond by day 9 (Fig. 4a, b) to ultimately reach 50 to >100 μm at the end point (Fig. 1c).

## Oncogenic EGFR status defines the molecular landscape of blood vessel-associated tumour microregions

To explore the molecular realm of these vascular changes, MES-GSC xenografts positive or negative for EGFR expression were subjected to digital spatial mRNA profiling with a focus on CD31$^+$ cells associated with large or small vessels in isogenic intracranial tumours. Vasculature in these xenografts was also compared to that in normal mouse brains. Formalin-fixed paraffin-embedded tumour sections (8 lesions) were used to select 42 regions of interest (ROI) (Supplementary Fig. 9a) followed by mRNA sequencing using Illumina NovaSeq platform and processing by GeoMx NGS Pipeline (DND) (NanoString; Supplementary Fig. 9b, c). While ROI-associated transcriptomes of EGFR-positive-, EGFR-negative tumour bearing and control brains revealed several differentially expressed genes (DEGs) between large and small blood vessels, their expression was only partially restricted to endothelial cells and may reveal more global changes in the perivascular microenvironment in EGFR-driven tumours (Supplementary Fig. S9d–f). A quantitative comparison of transcripts associated with large vessels from *EGFR*-WT tumours (vasectasia) versus small vessels from *EGFR*-KO tumours (angiogenesis), as depicted in the volcano plot (Supplementary Fig. 9e), demonstrated several significant differences (>2 fold; $p < 0.05$) with large vessels microregions being enriched for *Cd99*, *Pcbd2* and *Ripply2* relative to small vessel microregions. The functional aspects of this enrichment were also apparent from GSEA analysis of the respective molecular pathways, which pointed to upregulation of interferon response, reduced angiogenesis signal and expression of several vascular markers in ROIs associated with large vessels of EGFR-proficient tumours (Supplementary Fig. 10). We also compared the spatial gene expression profiles of GeoMX captured microregions surrounding both large and small vessels in GSC-MES xenografts and in the adjacent normal brain vasculature of comparable calibre. Those patterns were analysed against the recently described ATLAS of vascular markers associated with the brain arteriovenous axis[33]. Strikingly, while genes expressed in normal brain vessels in our study (large and small) mapped to some extent with the corresponding calibre vessels described in the ATLAS, tumour-associated regions containing blood

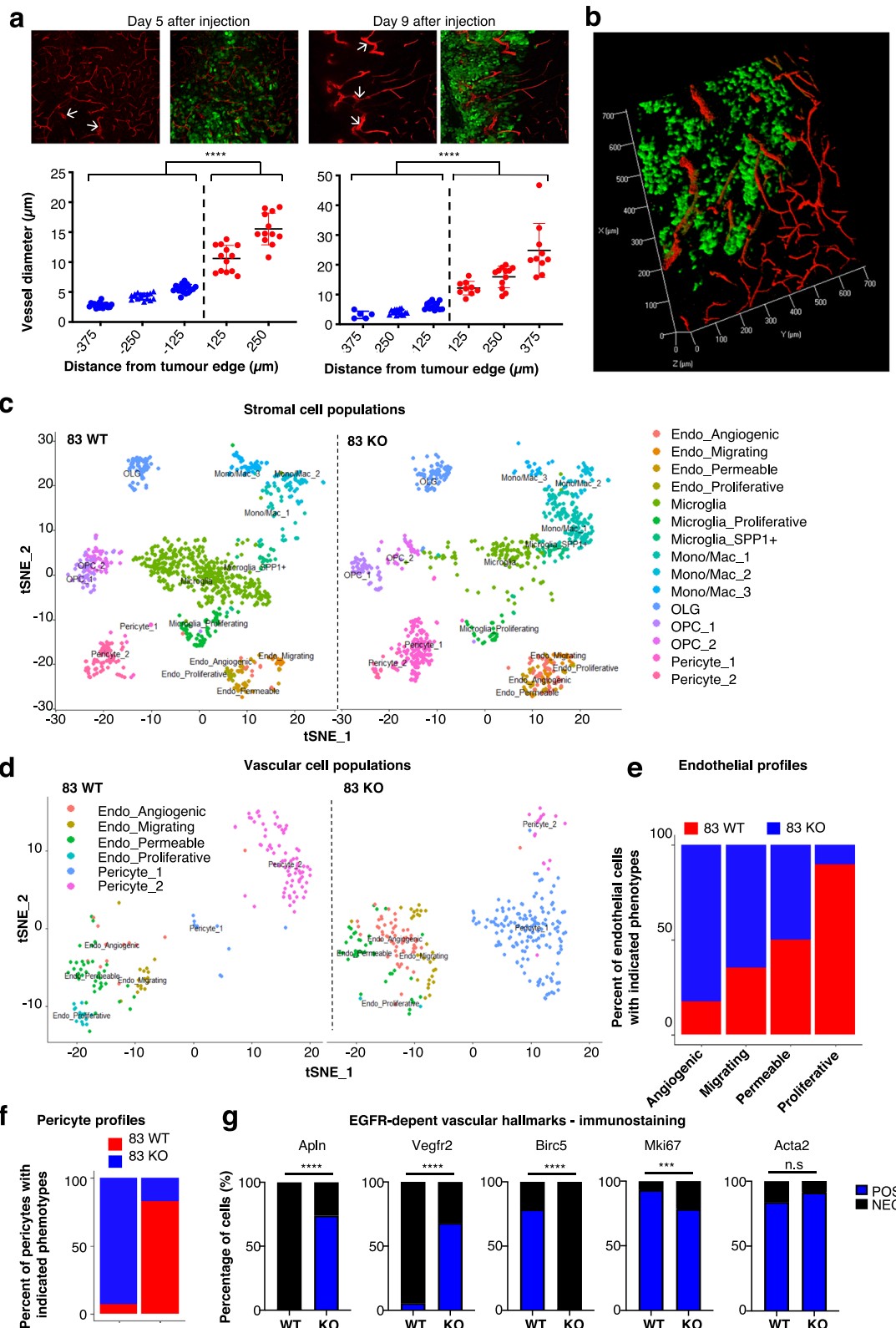

vessels of comparable sizes were markedly different (Supplementary Fig. 11a) suggesting a disease-specific imposition of abnormal regulatory influences, including both angiogenesis and vasectasia.

**Molecular traits of endothelial cells associated with vasectasia**

To further explore the molecular make-up of endothelial cell populations involved in vasectasia we subjected MES-GSC xenografts positive or negative for EGFR expression to single-cell transcriptional profiling (scRNAseq). The resulting subpopulations of cancer, stromal and endothelial cells were resolved computationally and compared to data reported in the literature[34]. Interestingly, GSC tumour cells, clustered separately, revealing distinct cellular landscapes (Supplementary Fig. 12a, b) dependent on the EGFR status. Similarly, we compared the clusters of murine (stromal) cells within both EGFR-proficient and

**Fig. 4 | Single cells sequencing to profile transcriptomes of vascular structures in human xenograft brain tumours. a** Enlargement of blood vessels diameter within the GSC83 tumour mass was detected 5 and 9 days after injections. Quantification of vessel size distribution was enabled by CD31 staining of endothelial cells ($n = 3$ independent experiments; two-tailed paired $t$ test $P = 4.22^{-25}$ and $2.83^{-14}$; Data are presented as mean values ± S.D.); **b** 3D reconstruction of GSC83 (83) tumour xenograft shows enlargement of blood vessels upon entry into the tumour. **c** T-distributed stochastic neighbour embedding (tSNE) plot shows clustering of murine cells based on gene expression. In both EGFR-KO and EGFR-WT GSC83 tumours cell colour specifies the assignment of cells to one of 15 different clusters inferred using shared nearest neighbour clustering; **d** tSNE plot of murine endothelial cells and pericytes shows clustering of cells based on gene expression. Two different populations of pericyte are detected and four different subpopulations of endothelial cells are identified, including: angiogenic, migrating, permeable, and proliferative; **e** Relative proportion of endothelial cell subpopulations in either EGFR-KO or EGFR-WT GSC83 tumours. **f** Relative proportion of pericyte cell populations in either EGFR-KO or EGFR-WT GSC83 tumours. **g** Quantification of immunostaining for vascular markers. Birc5 is selectively elevated in endothelial cells associated with vasectasia in EGFR-WT tumours, while Apln and Vegfr2 were upregulated in angiogenic blood vessels in EGFR-KO tumours ($n = 3$ independent experiments; two-tailed paired $t$ test $P = 6.60^{-23}-4.44^{-06}-1.84^{-16}$ and 0.00041); Source data are provided as a Source Data file.

-deficient tumours, which were biologically annotated based on the relative abundance of top-ranking marker genes (Fig. 4c and Supplementary Fig. 12c, d). In this regard, populations of microglia, monocyte/macrophages, oligodendrocytes with their precursors (OPCs), pericytes and endothelial cells were readily distinguished and comparable, but not identical between tumours with intact or disrupted *EGFR* gene (Supplementary Fig. 12e, f).

Interestingly, single-cell gene expression suggested the existence of at least four subpopulations of endothelial cells classified as migrating, proliferative, permeable and angiogenic. Two distinct subsets of pericytes were also detected (Fig. 4d–f; Supplementary Fig. 13a, b). Notably, EGFR-expressing tumours were enriched in proliferative endothelial cells, while *EGFR*-KO tumours contained mostly angiogenic endothelia (Fig. 4e).

Single-cell gene expression data also pointed to human *EGFR* transcripts being detectable in endothelial cells of both human brain tumours and murine brain xenografts (Supplementary Fig. 13c–e). Thus, in MES-GSC xenografts, subpopulations of CD31⁺ and CD34⁺ mouse endothelial cells harboured human *EGFR* transcript (Supplementary Fig. 13d). Similarly, in silico analysis of scRNAseq human GBM datasets (GSE84465) also revealed the presence of *EGFR* transcripts in CD34⁺ endothelial cells (Supplementary Fig. 13c). In addition, the presence of human EGFR protein was detected by flow cytometry in CD31-postive endothelial cells isolated from EGFR-WT xenografts (>15 fold enrichment; $p < 0.001$). Moreover, using magnetic bead separation (MACS) of EGFR-positive endothelial cells, we were able to detect their population containing human phosphorylated EGFR (Supplementary Fig. 13e, f). While a relatively small fraction of endothelial cells exhibited this phenotype, their sustained presence is consistent with in vitro results suggesting a role of activated endothelial EGFR in vasectasia. Finally, heterogeneous populations of microglia and myeloid cells were also detected and varied between tumours with different EGFR status (Supplementary Fig. 14a–e), while oligodendrocytes and their precursors (OPCs) did not exhibit major differences in this regard (Supplementary Fig. 14f).

### Molecular traits associated with glioma stem cell-driven neo-vascularization patterns

A more in-depth examination of differentially expressed transcripts between EGFR-expressing and -non-expressing MES-GSC driven tumours additionally exposed specific molecular distinctions between vascular patterns dominated by either angiogenesis or vesectasia. As predicted from scRNAseq profiles, tissue immunostaining revealed that protein markers of endothelial tip cells associated with capillary sprouting outgrowth, such as Apln (apelin) and Vegfr2 (VEGF receptor 2), were widely expressed among CD31+ cells in MES-GSC83-KO tumours (with disrupted *EGFR* gene expression). In contrast, larger vessels from EGFR-expressing tumours were enriched for Birc2 (survivin), Socs2 (suppressor of cytokine signalling 2) and Srsf2 (serine and arginine-rich splicing factor 2). In addition, the presence of Ki67 staining not only in angiogenic (KO-EGFR), but also in vasectasia-related (EGFR-WT) blood vessels implicated the ongoing endothelial proliferation, as a part of both processes (Fig. 4g, Supplementary

Fig. 15a). Finally, RNAscope in situ hybridisation (ISH) analysis further confirmed the presence of distinct transcripts (Socs2, Srsf2, Birc2) in endothelial cells associated with MES-GSC-driven and EGFR expressing tumours (Supplementary Fig. 15b).

### Targeting EGFR disrupts vasectasia and overcomes resistance to anti-VEGF therapy

Antiangiogenic VEGF-directed therapies have not improved overall survival in unstratified GBM patient populations[1] likely due to alternative and unrecognised mechanisms of tumour neovascularization[17]. Since aggressive MES-GSCs express mediators of both angiogenesis (VEGF) and vasectesia (EGFR-EVs; Fig. 1i–k), we considered the consequences of targeting both of these respective processes in vivo, either separately or simultaneously (Fig. 5). To this end, NSG mice were intracranially inoculated with MES-GSC cells (GSC83) expressing Luciferase. Once the bioluminescent tumour signal became apparent, the animals were randomised to receive anti-mouse VEGFR2 blocking antibody (DC101), EGFR inhibitor (DAC), both, or vehicle controls (IgG, Lactate; Fig. 5a; Supplementary Fig. 16a, b;). Interestingly, the administration of DC101 delayed tumour growth somewhat, but this effect wore off within 2–3 weeks and tumours resumed rapid growth trajectory. DAC therapy inhibited GSC83 tumour progression (in spite of the resistance of these cells to DAC treatment in vitro—Fig. 3d), but prolongation of survival was modest. Strikingly, the symptom-free survival of mice receiving both agents was markedly extended and reached approximately 30 days beyond the baseline (3-fold extension; Fig. 5b; Supplementary Fig. 17a).

These responses were associated with remarkable rearrangements of vascular patterns (Fig. 5c–e). Thus, vasectasia, a network of mostly larger vessels was observed in control tumours and following DC101 treatment alone. In contrast, tumours exposed to DAC were essentially devoid of blood vessels with diameters greater than 50 μm (hallmark of vasectasia), but exhibited dramatically elevated microvessel density, possibly due to compensatory angiogenesis Fig. 5d, e). The latter response was attenuated when DAC was combined with DC101 (Fig. 5e) suggesting a role for an interplay between EGFR and VEGF/VEGFR2 pathways in microvascular growth processes. Overall, these results are consistent with the notion that MES-GSC-initiated brain tumours mount multiple vascular growth responses, either through vasectasia driven by EGFR (EGFR-EVs) or through angiogenesis, the latter dependent, at least in part, on the VEGF pathway.

## Discussion

Overall, our study brings several important elements into the ongoing effort to understand the nature, role and therapeutic opportunities associated with GBM neovascularization. Disappointing experiences with antiangiogenic agents[1] triggered a renewed interest in non-angiogenic vascular processes in GBM, such as perivascular invasion, vascular mimicry and cooption of pre-existing blood vessels by brain cancer cells[17,35]. While the latter mechanism is increasingly well documented[36], the vascular architecture of GBM does not resemble that of normal brain and endothelial cell proliferation is frequently observed[2], suggesting the involvement of active vascular growth

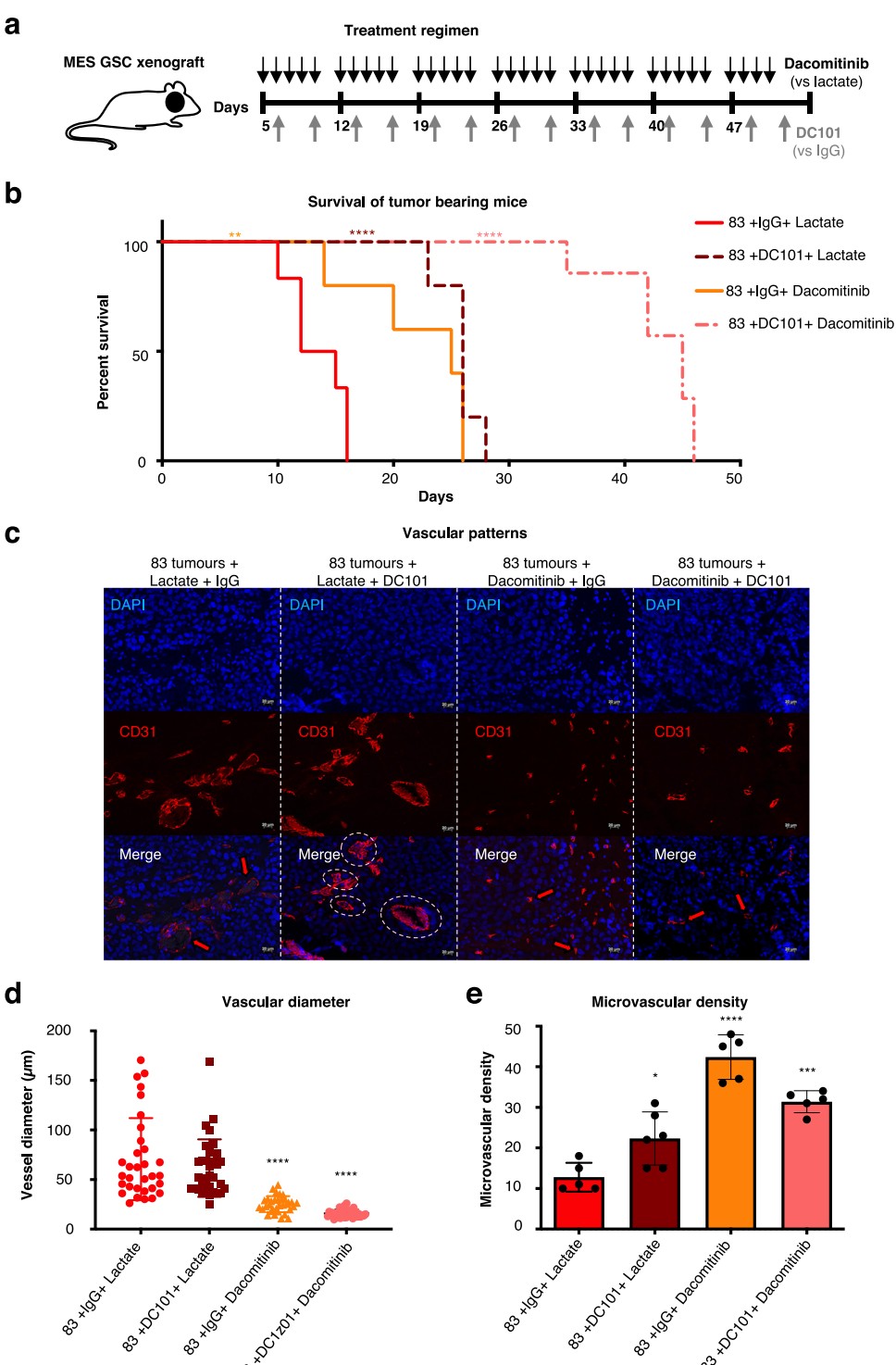

processes (rather than passive cooption), including and beyond angiogenesis. Our study suggests that at least one such process may involve circumferential, rather than angiogenic, vascular growth (vasectasia) driven largely (perhaps not exclusively) by EV-mediated transfer of EGFR between tumour and endothelial cells.

Circumferential enlargement of blood vessels could be driven by several mechanisms. For example, at specific stages of development overexpression of angiopoietin 1 may disbalance processes of endothelial proliferation and outgrowth leading to formation of larger vessels[37]. Vessel enlargement may also result from

deficiencies in Id1/Id3 genes[38] or VEGF-dependent overactivation of the NOTCH pathway impacting the recruitment of tip cells to endothelial sprouts[18]. Deregulation of SLIT2-ROBO signalling in microglia may also have similar effects[39]. These mechanisms may contribute to abnormal vascular patterns in brain tumours[18,39] but their linkage to molecular drivers that define GBM subtypes[6] is presently unclear and relatively unstudied. Intriguingly, mesenchymal GBMs appear to be enriched in larger calibre blood vessels[40]. However, cellular heterogeneity of GBM cell populations and the coexistence of several subtypes of GSCs in different tumour

**Fig. 5 | Combined targeting of VEGFR2 and EGFR suppresses tumour growth and vasectasia. a** Schematic representation of the treatment with EGFR and VEGFR-2 inhibitors. Dacomitinib was administered by gavage at 15 mg/kg for 5 consecutive days followed by a 2-day break, while DC-101 was injected intraperitoneally at 20 mg/kg, twice a week. Lactate and IgG at corresponding concentrations were injected in parallel, as placebo controls. The mice were treated with single agents alone, or in combination; **b** Survival curves of mice harbouring mesenchymal glioma stem cell xenografts (GSC83) subjected to a combination therapy targeting VEGF and EGFR pathways. Kaplan-Meier plot depicts groups of mice treated with placebo (IgG + lactate), DC-101 or Dacomitinib alone or with combination therapy of DC-101+ Dacomitinib ($n = 5$ independent experiments; two-tailed paired $t$ test; $P = 0.0075$–$0.000051$ and $0.0000223$); **c** Representative images of immunofluorescent staining for CD31 reveals the impact of therapy on vascular patterns. **d** Quantification of vessel size distribution based on CD31 staining. Blood vessels in mice treated with placebo (IgG + lactate) and single treatment with DC-101 present enlarged lumen compared to a mean vessel diameter of GSC83 (83) tumours treated with Dacomitinib and combination therapy of DC-101+ Dacomitinib ($n = 5$ independent experiments; two-tailed paired $t$ test; $P = 4.99^{-08}$ and $2.18^{-10}$); **e** Quantification of microvascular density following CD31 staining. Microvascular density was increased in tumours treated with Dacomitinib and combination therapy of DC-101+ Dacomitinib as measured by vessel numbers per high power field (hpf; $n = 5$ independent experiments; two-tailed paired $t$ test; $P = 7.92^{-06}$ and $1.45^{-05}$). Data were presented as means ± SD. Significance: **$p < 0.01$, ***$p < 0.001$, ****$p < 0.0001$ treated group versus untreated control group; Source data are provided as a Source Data file.

regions[41] may lead to complex and regional vascular patters combining several mechanisms of neovascularization.

Our observations suggest a link between the expression of EGFR by the mesenchymal (but not proneural) subset of tumour-initiating GSCs and their ability to orchestrate formation of vascular patterns enriched in large intratumoural blood vessels and low overall vascular density. We also suggest that at least one mechanism triggering these changes involves the EV-mediated transmission of EGFR/EGFRvIII from cancer cells to endothelium and prolonged ectopic activation of this oncogenic receptor in these cells. Due to these distinctive and cancer-specific features, we refer to this process as *vasectasia* to distinguish it from both angiogenesis and other forms of blood vessel enlargement.

It is interesting to note that efficient intercellular transmission of the functional EGFR transcript by exosome-like small EVs may be crucial (and sufficient) to enable a prolonged-expression and activation of EGFR in endothelial cells. While the mechanisms of RNA packaging into EVs remain incompletely elucidated and appear to favour small RNA bioforms[42], EV-mediated intercellular transmission of functional transcripts has also been reported[23]. Indeed, our preliminary and ongoing analysis suggests that cargo of MES-GSC EVs contains both EGFR mRNA and multiple RNA binding proteins capable of interacting with this transcript[26].

EVs are also capable of intercellular shuttling of regulatory surface proteins, including EGFR kinase[20,43,44]. While these events are functionally meaningful the half-lives of ectopic proteins in recipient cells could be more limited, due to rapid turnover, and thus their pools could be increased by concomitant transfer and translation of mRNA. Interestingly, different subsets of MES-GSC EVs appear to carry EGFR protein or EGFR transcript, and this duality may influence processes of their intercellular transfer and biological activity in ways that remain to be elucidated.

Finally, our observations suggest that vasectasia represents a targetable process through EGFR inactivation and possibly through interference with its endothelial mediators and EV carriers. Interestingly, EGFR targeting appears to be able to overcome, at least partially, the acquired resistance of glioblastoma xenografts to inhibition of VEGF-driven angiogenesis. Whether this approach would synergise with the standard of care in GBM remains to be studied. Equally important is the larger picture of regional heterogeneity and micro-environmental complexity of GBM, as well as stress factors, such as hypoxia and therapy as an important context for vasectasia. We suggest that a better understanding of EV-mediated processes within the GBM microenvironment may lead to new forms of precision therapy in this dismal disease.

## Methods
### Mice
All procedures involving animals were performed in accordance with the guidelines of the Canadian Council of Animal Care (CCAC) and the Animal Utilization Protocols (AUP) approved by the Institutional Animal Care Committee (ACC) at RI MUHC and McGill University (Protocol #5200). The NSG (NOD scid IL2Rgamma-/-) transgenic mice were injected intracranially with GSCs (25,000 cells/μL with total volume of 2 μl) using a Stoelting Stereotaxic Injector at pre-determined coordinates (2.5-1.5-3.0) of bregma and sagittal suture as described[3]. Additional details can be found in Supplementary Information file.

### Cell culture conditions
Glioma stem cell (GSC) lines were isolated in the laboratory of Dr. Ichiro Nakano (University of Alabama at Birmingham, AL). The cell lines of either proneural (GSC157; GSC1079; GSC528) or mesenchymal (GSC83; GSC1005) subtype were developed in the form of sphere cultures isolated from surgical samples of glioblastoma (GBM) patients. Both types of GSCs were maintained as spheres in the medium containing DMEM-F12 (GIBCO, Catalog No. 11320033) supplemented with 100 μg/ml EGF (GIBCO, Catalogue No. PHG0311L), 100 μg/ml FGF (GIBCO, Catalogue No. PHG0261), 0.2% Heparin (STEMCELL, Catalogue No. 07980), 1X B27 serum-free supplement (GIBCO, Catalogue No. 17504044), 1% Glutamax (GIBCO, Catalogue No. 35050061) and 1% penicillin-streptomycin (P/S) (GIBCO, Catalogue No. 15070063). EOMA (ATCC, Catalogue No. CRL-2586), a mouse hemangioendothelioma (transformed endothelial) cells, and HBMVEC (iXCells Biotechnologies, Catalogue No. 10H-051), human brain microvascular endothelial cells, were cultured on 0.1% gelatin-coated plates. Cells were maintained in DMEM-F12 medium supplemented with 10% FBS, 1% P/S and 40 μg/mL endothelial growth supplement (ECGS) (Sigma, Catalogue No. E2759). HUVEC are normal human primary umbilical vein endothelial cells, commercially available (ATCC, Catalogue No. PCS-100-010) that were cultured on 0.1% gelatin-coated plates using EGM-2 Bullet-Kit media (Lonza, Catalogue No. CC-3162).

### Extracellular vesicle isolation method
EVs were purified by differential centrifugation (Beckman TLA100.2 rotor) from the indicated conditioned media of monolayer cell cultures. After cell debris was eliminated by centrifugation at $2000 \times g$ for 20 min, the supernatant was concentrated (centrifuged at $3500 \times g$ for 20 min) using Amicon Ultra-15 Centrifugal Filter Units −100 kDa- (Millipore # UFC905008) to a final volume of 1 mL. The concentrated conditioned medium was passed through 0.22 μm filter and then centrifuged at $110,000 \times g$ for 70 min. The resulting EV pellet was re-suspended in filtered 1 × PBS or RIPA buffer and stored at −80 °C.

### RNA extraction and RT-PCR and gel electrophoresis
Total RNA was extracted from cells or EVs using TRIzol Reagent (Invitrogen # 15596026) and RNeasy Mini Kit (Qiagen # 74104, Mississauga, ON, Canada) according to manufacturer's recommendations. The cDNA obtained was then amplified using human EGFR and EGFRvIII primers (see Supplementary information for details) and PCR was performed on 2.5 μL of prepared cDNA using MyTaq Red DNA Polymerase (Bioline, London, UK). Amplified PCR products were resolved on 2% agarose gel for 30 min at 100 V and the DNA bands

were visualised using ultraviolet (UV) transilluminator gel documentation system.

## Nanoparticle tracking analysis (NTA)

Extracellular vesicle size and quantity were analysed using NS500 (Nanosight; Malvern Panalytical, Malvern, UK) NTA instrument. NanoSight relies on light scattering to visualise particles in the range of 100 nm to 2 μm, records their movement in 30 s video files, tracks individual particles and calculates concentration and size based on Brownian motions. The samples were diluted with D-PBS to reach optimal loading concentration of $10^7$–$10^9$ particles per mL. Analysis was performed as described earlier[1].

## Protein quantification and western blot (WB)

Total proteins from cells were extracted using RIPA buffer containing 7 × protease inhibitor (Roche, Catalogue No. 11836153001), and solubilized proteins were quantified using the Pierce Micro BCA™ Protein Assay (Thermo Scientific, Rockford, IL, USA). Proteins were resolved using sodium dodecyl sulfate–polyacrylamide gel electrophoresis (SDS- PAGE), at 10%, and transferred to polyvinylidene difluoride membranes (PVDF; Biorad, Mississauga, ON, Canada). The membranes were probed with indicated primary antibodies, and appropriate horseradish peroxidise-conjugated secondary anti-mouse (Biorad # 170-6516), or anti-rabbit (Cell Signaling # 7074S) antibodies. Amersham ECL Western Blotting Detection Kit (RPN2108 GE Healthcare) was used for the detection of chemiluminescence and band visualisation using ChemiDoc MP system (Biorad).

## Cell growth/survival assays (MTS Assay)

Cell titre 96 (Promega # 43580) assay was used to measure in vitro cell growth/viability in the presence of Dacomitinib or EGFR knockout. As indicated, $7 \times 10^3$ GSC cells/well were seeded in 96 well plates in complete growth media for 24 h.The following day the cells were washed and treated with 2.5 μM Dacomitinib in DMEM containing 1% FBS or control media. For EGFR-KO and EGFR-WT analysis the cells were left in complete growth media for the duration of the assay. The absorbance at 490 nm was read at time intervals indicated and the signal reflective of viable cell numbers was assessed for up to 6 days.

## Transwell migration assay

Gelatin (0.1%)-coated 8.0 μm transwell inserts were placed in 24-well plates, and HUVEC cells ($2 \times 10^3$) were plated into the inserts. The following day, HUVECs were washed with PBS twice and starvation media with 1% EV-depleted FBS[2] was added to the cells, and to the lower part of the well. Conditioned media (1:1), EV-depleted supernatant (1:1), or EVs (30 μg/mL), from each cell line, versus controls containing buffer (PBS) with no EVs, were added onto the HUVEC cells to stimulate their migration. After incubation for 3 days, inserts containing cells were fixed with 3.7% formaldehyde, washed with PBS, which was followed by staining with 0.5% crystal violet solution. Finally, the inserts were examined under the light microscope and quantification of migrated cells was performed using FIJI software.

## Immunofluorescent staining (IF)

Tumour tissues were preserved in 4% paraformaldehyde (PFA) immediately after resection from mice. They were then run through a series of automated processing steps executed in a Leica TP 1050 tissue processor. The resulting paraffin-embedded blocks were sectioned using American Optical microtome into 4 μm thick tissue sections and placed on slides. Sections were re-hydrated, antigen retrieval was performed in 0.01 M citrate buffer (pH 6), followed by blocking with PBS containing 5% serum and staining with primary antibodies (see Supplementary informatioin for details). Mounting solution, Vecta-Shield® HardSet™ with DAPI, was used to seal the slides with coverslip.

## Lentivirus production

VSV-G (8454 Addgene), pRRE (12251 Addgene), REV (12253 Addgene) and transfer plasmids were added to $4.5 \times 10^6$ 293 T cells. As transfer plasmids, we used sgRNAs in pCLIP-Dual-SFFV-ZsGreen for EGFR CRISPR guides (TEDH-1024003, TEDH-1024000, TEDH-1024001, TEDH-1055978 Transomic) and pCLIP-Cas9-Nuclease-hCMV-tRFP (SHB_2264 Transomic) for the CAS9. Guide RNA plasmids and CAS9 plasmids were obtained from Dr. Sidong Huang, McGill University. For generating luciferase-positive cells, we used the previously described pSMAL vector modified from the MA1 lentiviral vector to have a Gateway cassette and SFFV promoter (PMID: 15619618 and PMID: 24776803) and with luciferase gene cloned from pGL4.51(luc2/CMV/Neo) (E1320 Promega) (kindly obtained from Dr. K. Eppert, McGill University). Finally, the pellets of viral particles were spun at 22000 rpm for 2 h. Obtained pellets were re-suspended in 50 μL of PBS. pSMAL lentiviral vectors were used to transduce MES-GSC.

## BME plug vascular growth assay

Cold liquid growth factor-reduced Cultrex basement membrane extract (BME) solution (3433-010-R1 R&D system) was mixed with 100 μg of EVs or VEGF at indicated concentrations, injected subcutaneously into C57BL/6 mice and allowed to solidify to form a palpable pellet. Pellets were collected on day 21 post injection, photographed, imaged by microscope and placed in sucrose for cryopreservation and histology.

## Aortic ring assay

Aortas of 4-week-old C57BL/6 mice were isolated and the rings were cultured in growth factor-reduced Cultrex BME (3433-010-R1 R&D system) polymerised at 37 °C. The rings were observed until sprout-like endothelial outgrowths started to appear after which they were placed in 1% FBS supplemented with either 30 μg/mL of EVs, VEGF or vehicle. The number of outgrowths was quantified using images, which were analysed using the Fiji distribution of ImageJ (PMID: 22743772) with Angiogenesis Analyser plugin.

## VEGF Elisa

For the detection of VEGF secreted from GSC cells, either as soluble factor in the supernatant or released in EVs, we employed ELISA kit purchased from R&D Systems (#RRV00) and used it according to manufacturer's protocol.

## Phosphorylated receptor tyrosine kinase antibody array

To explore the intracellular signalling mechanisms triggered in HUVEC or HBEC-5i cells by cancer EVs, 30 μg/mL of GSC-derived EVs were combined with endothelial cells and incubated for six days. The cells were collected, lysed in RIPA buffer and the relative expression of phosphorylated kinases was analysed using Human Phospho-RTK Array kit and the Human Phospho-MAPK Array kit (both from R&D Systems, Minneapolis, MN, USA), followed by quantification using Fiji software.

## Immunoprecipitation of EGFR-positive EVs

Dynabeads™ Protein G was incubated overnight with rabbit anti-EGFR antibody (4267 Cell signaling). The day after the antibody was removed, beads were washed three times in PBS and incubated overnight with intact EVs (30 μg). Using a magnet, beads coated with EVs enriched for EGFR protein were separated from the flow-through fraction. The EGFR-positive EVs and EGFR-negative EVs were then lysed either in RIPA buffer or in Lysis buffer for RNA extraction (as mentioned above).

## Lycopersicon lectin injection for vascular imaging

Lycopersicon lectin (DL-1178, vector Laboratories) was injected i.v. 30 min prior to humane euthanasia of the mice. Brains from mice were

harvested and immersed in cold PBS to be sectioned at 200 μm thickness using a vibratome (Leica VT 1200 s). The tissues were placed in a μ-Dish 35 mm, high Glass Bottom dish (81158, ibidi) and subjected to high-resolution confocal microscopy (Zeiss LSM780 laser scanning confocal microscope).

## Single cells sequencing and data analysis

Whole-tumour specimens were dissociated and the cells isolated using Collagenase/Dispase (11097113001 Millipore) to be resuspended in PBS for single-cell capture. Following the Chromium Single Cell 3′ Reagent Kits v3 User Guide (CG0052 10x Genomics)[5], a single-cell RNA library was generated using the GemCode Single-Cell Instrument (10x Genomics, Pleasanton, CA, USA). The sequencing-ready library was purified with SPRIselect, quality controlled for sized distribution and yield (LabChip GX Perkin Elmer) and quantified using qPCR (KAPA Biosystems Library Quantification Kit for Illumina platforms P/N KK4824). We used Cell Ranger v3.0.1 (10x Genomics) to demultiplex the raw sequencing reads to FASTQ files and align the reads to human and mouse reference hg19 and mm10 to quantify gene counts for each origin of species (UMIs), getting about 349 million and 404 million read counts for EGFR-WT and -KO samples, respectively. We loaded the gene count data using the Seurat pipeline and the normalised data was visualised using the Seurat and dittoSeq packages.

## Data collection and statistical analysis

All experiments were reproduced at least 2–3 times with similar results unless otherwise indicated. Statistical analysis was carried out using a computerised two-tailed Student's $t$ test and ANOVA. Error bars represent standard deviation. Asterisks indicate: * - significance at <0.05, ** - significance at <0.01, *** - significance at <0.001 and **** - significance at <0.0001. Please see Supplemental Material and Methods for further experimental detail[10,20,45-47].

## Reporting summary

Further information on research design is available in the Nature Portfolio Reporting Summary linked to this article.

## Data availability

The single-cell sequencing and spatial whole transcriptome digital spatial profiling (GeoMX) data generated in this study have been deposited as unfiltered and filtered R objects as well as raw data in the Gene Expression Omnibus (GEO) database under accession code GSE207360. For the whole transcriptome digital spatial profiling data, the data are also available as a supplement to the manuscript in the formats of raw expression (DCC and PKC files) and processed expression (filtered for targets detected in at least 5% of the ROIs and Q3- normalised) data along with the annotation file for each ROI. Source data for main and Supplementary Figs. are provided as a separate file along with the corresponding Source Data file. Code availability: The code used to produce the results of scRNA-seq analysis is available at https://github.com/mera3113/Vasectasia. Source data are provided with this paper.

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

## Acknowledgements

This work was supported by grants from Canadian Institutes for Health Research (PJT183971), as well as gifts from Fondation Charles Bruneau and Fondation CIBC (NET Programme) to J.R. who is also a recipient of the Jack Cole Chair in Paediatric Haematology/Oncology. Y.R. is a recipient of funding from KFOC and Department of Defence (DoD), US. Infrastructure funds were provided by Canada Foundation of Innovation CAN programme to JR and by Fonds de Recherche en Santé du Quebec (FRSQ) to RIMUHC. C.S. and L.A. were supported by FRQS fellowships. We are grateful for helpful suggestions from our colleagues and the support of our families.

## Author contributions

Contributions: C.S. and J.R., developed the study concept and design; C.S., L.A., B.M., L.M. and T.N. conducted experimental work and analysed the data. I.N. provided crucial reagents and guidance. S.E.C. and S.H. provided samples; M.K. provided datasets and bioinformatic analysis; T.N. and Y.R. performed GeoMX experiment. C.S. and J.R. wrote the original draft; and all authors revised and approved the manuscript before submission.

## Competing interests

The authors declare no competing interests.
