## [Peer Review File · Nature Communications]

REVIEWER COMMENTS

Reviewer #1 (Remarks to the Author):

This thorough and rigorous paper addresses mechanisms of angiogenesis in glioblastoma, contrasting proneural and mesenchymal angiogenesis. The paper elucidates potential reasons for failure of anti-VEGF glioblastoma therapy. It also provides compelling evidence from cell and PDX models for the role of extracellular RNA communication and Extracellular Vesicles (EVs) in cancer progression. Spatial single-cell profiling of PDX models integrates tissue-level and molecular perspective. Dual targeting of EGFR and VEGF points at specific combination therapies that may need to be applied in clinical trials. Overall, the paper is comprehensive, solid and of potentially high interest to Nature Communications audience. A number of minor issues and requests for clarification are listed below.

1. Fig 1f – The sprouting for MES exceeds that of PN, which is not concordant with higher microvascular density of PN vs MES illustrated in 1d.
2. Fig 1i and j, also Supp F1c, suggest MES EVs stimulate VEGF-dependent sprouting without VEGF protein being detectable within them. This should be addressed. Are there mRNAs, miRNAs or other noncoding RNAs in these EVs that may explain this activity?
3. Fig2b – What is the positive control for the effect of PN EVs on endothelial cells?
4. Fig3c – VEGF is shown but legend says EGFR only. Moreover, the figure shows that MES EVs promote sprouting and not vasectasia, as would be expected from discussions elsewhere in the paper.
5. Fig3d – In light of emphasis of the paper on EGFR effect on vasectasia, in contrast to angiogenesis, why was vasectasia not assayed? Why would the effect of EGFR on migration be more related to vasectasia than angiogenesis?
6. scRNA-seq of EGFR+ and EGFR- xenografts reveals distinct EGFR+ and EGFR- clusters with no cluster spanning EGFR+ and EGFR-. This pattern is highly conspicuous and may possibly be explained by a batch effect due to separate handling and sequencing of EGFR+ and EGFR- tumors. For the current explanation to stand, such batch effect needs to be ruled out.
7. 5e,g: It is not clear how 83 WT and 83 KO add up to 100%. How are the percentages specifically calculated?
8. Lines 269-270: The sentence may be mis-interpreted to mean that angiogenesis is just “vessel enlargement”. Why is vasectasia not part (or an aspect of) angiogenesis?

Reviewer #2 (Remarks to the Author):

Mesenchymal glioma stem cells trigger vasectasia, a distinct neovascularization process mediated by extracellular vesicles carrying EGFR

C. Spinelli , L. Adnani ,B. Meehan , L. Montermini , S. Huang, M. Kim, T. Nishimura, S.E. Croul; Y. Riazalhosseini and J. Rak

Manuscript Peer-Review Report:

Glioblastoma multiforme is the most deadly form of malignant tumor rendering a median patient survival period of under 15 months despite undertaking currently available treatments¹. Transcriptional profiling has allowed the classification of GBM tumors into four subtypes: classical, mesenchymal, proneural and neural, stemming from glioma stem cells (GSCs), the GBM tumor initiating cells divided into proneural (PN) and mesenchymal (MES) subtypes, contributing to the heterogeneity in GBM². The differences in transcriptional expression of the epidermal growth factor receptor (EGFR) observed in GBM has been linked to the aggressiveness of GBM tumors presumably due to its role in cell proliferation, migration, angiogenesis and survival^{3,4}. Therefore, a better understanding of the role of EGFR in GBM would be beneficial towards the development of GBM treatments.

In the manuscript under review, Rak et al demonstrated that in comparison to PN-GSC xenografts, the stunted survival of tumor bearing mice with MES-GSC xenografts was related to the differences in vascular patterning resulting from EGFR expression in MES-GSC extracellular vesicles (EVs). The presence of pEGFR specific to the MES-GSCs was deemed responsible for the formation of a less dense vasculature with larger vessel formation as observed through fluorescence imaging of Anti-CD31 in endothelial cells. Since mouse aortic rings stimulated with culture media from GSC cultures and with pelleted EVs (isolated through centrifugation of corresponding culture media) demonstrated an increased sprouting response in both cases, further experiments were performed to decipher what factors are responsible for the sprout induction in both circumstances. One such factor was discovered to be VEGF but its expression remained exclusive to culture media and absent in EVs. Therefore, further experiments were conducted to assess what factors in EVs are responsible for the development of this effect. Therein, the phosphorylation of EGFR was detected in endothelial cells stimulated with EVs from MES-GSCs and Western Blot analysis demonstrated that EGFR (and the GBM-specific mutant EGFRvIII) protein and mRNA expression may be transferrable from EVs to endothelial cells. Negative controls such as the EGFR inhibitor Dacomitinib and EGFR/EGFRvIII gene knock-out in MES-GSCs resulted in a decrease in endothelial cell migration, sprouting, increased the survival of tumor bearing mice and reverted vasculature changes thereby proving the importance of EGFR expression in MES-GSC EVs responsible for

the observed tumorigenic enhancement in this GSC subtype. Additionally, single cell transcriptional profiling was conducted for MES-GSC xenografts positive and negative for EGFR and these results classified endothelial cells into migrating, proliferative, permeable and angiogenic subpopulations. Finally, the top 100 genes variably expressed between the large and small vessels of EGFR-WT tumors, EGFR-KO tumors as well as control brain tissue were listed.

In conclusion, the authors described the existence of two different GSC-mediated blood vessel growth patterns: PN-GSCs that lack the presence of EGFR producing dense capillary networks through the mediation of angiogenesis and MES-GSCs that carry EGFR/EGFRvIII in EVs triggering 'vasectasia', a term they coined for the observed circumferential extension of tumor blood vessels. Correspondingly, the authors proposed a novel approach to GBM therapy by combinatorial targeting of VEGF/VEGFR2 and the EGFR pathway through the administration of the inhibitor DC101 and inhibitor Dacomitinib, respectively, and demonstrated its therapeutic potential by presenting a prolonged lifespan of MES-GSC xenograft tumor bearing mice simultaneously reverting their vasculature changes to the PN-GSC phenotype.

I believe that this manuscript is promising and the experiments are very well conducted. However, I have written a few suggestions below that would be beneficial to strengthen the premise in this paper.

Suggestions for improvement:

1. A study by Reardon et al from 2010 illustrated the outcomes of VEGF and EGFR combinatorial targeting in GBM patient clinical trials utilizing the administration of Bevacizumab with Erlotinib5. The outcome prognosis was not ameliorated in comparison to historical Bevacizumab consisting treatments. I recommend elaborating why Dacomitinib and DC101 treatment would be superior for consideration in GBM treatment in comparison to previously attempted combinatorial drug approaches.
2. J. Rak has previously published reports describing that GBM cells release EVs and that glioma cells release membrane-derived microvesicles consisting of oncogenic EGFRvIII6,7. I recommend elaborating why studying the phenomenon of EGFR presence in MES-GSC EVs is superior or significant for understanding GBM therapeutic approaches in comparison to already known effects of EGFR derived from microvesicles in GBM cells.

Minor suggestions:

1. More attention should be directed towards the use of proper units. E.g., micro should be written as 'μ' instead of 'u' as observed in lines 172, 318, 333, 376, 379 and Figure 4b.
2. In Figure 2, although the images demonstrate Western Blot results, there is no indication in the text as to how many times these experiments were repeated ('n').
3. Certain figure y-axis labels should include units of measurement or these units should be described in the text (e.g. Figure 4d).

Discretions:

1. Certain figures would benefit visually from improved symmetry and alignment of statistic indicators or axis labels (e.g. Figure 4b, d).
2. In certain experiments that portray a reduced specific effect after knocking-out the EGFR gene or use of inhibitor Dacomitinib (Figures 3b-d) it would be beneficial to demonstrate the statistics indicating the diminution in effect after obliterating EGFR function by comparing the results to the WT control rather than just comparing the values to the untreated control.
3. In the description for Figure 5, various upregulated genes are listed. It would be beneficial to the reader if the significance of these genes was briefly described. What are the roles of these genes? Are they in any way related to one another?

References:

1. Cloughesy, Timothy F et al. "Glioblastoma: from molecular pathology to targeted treatment." Annual review of pathology vol. 9 (2014): 1-25. doi:10.1146/annurev-pathol-011110-130324
2. Steponaitis, Giedrius, and Arimantas Tamasauskas. "Mesenchymal and Proneural Subtypes of Glioblastoma Disclose Branching Based on GSC Associated Signature." International journal of molecular sciences vol. 22,9 4964. 7 May. 2021, doi:10.3390/ijms22094964
3. Verhaak, Roel G W et al. "Integrated genomic analysis identifies clinically relevant subtypes of glioblastoma characterized by abnormalities in PDGFRA, IDH1, EGFR, and NF1." Cancer cell vol. 17,1 (2010): 98-110. doi:10.1016/j.ccr.2009.12.020
4. Xu, Bin et al. "MicroRNAs involved in the EGFR pathway in glioblastoma." Biomedicine & pharmacotherapy = Biomedecine & pharmacotherapie vol. 134 (2021): 111115. doi:10.1016/j.biopha.2020.111115
5. Sathornsumetee, Sith et al. "Phase II trial of bevacizumab and erlotinib in patients with recurrent malignant glioma." Neuro-oncology vol. 12,12 (2010): 1300-10. doi:10.1093/neuonc/noq099
6. Al-Nedawi, Khalid et al. "Intercellular transfer of the oncogenic receptor EGFRvIII by microvesicles derived from tumour cells." Nature cell biology vol. 10,5 (2008): 619-24. doi:10.1038/ncb1725
7. Choi, Dongsic et al. "The Impact of Oncogenic EGFRvIII on the Proteome of Extracellular Vesicles Released from Glioblastoma Cells." Molecular & cellular proteomics : MCP vol. 17,10 (2018): 1948-1964. doi:10.1074/mcp.RA118.000644

Reviewer #3 (Remarks to the Author):

The manuscript by Spinella et al focuses on angiogenesis versus vasectasia, the latter suggested to be induced by extracellular vesicles (EVs) from mesenchymal (MES) glioma stem cells (GSCs), and the role of EGFR/EGFRvIII in promoting this distinct vascular phenotype called vasectasia. Overall the manuscript is very interesting and contains novel data that are of physiologic relevance. However, there are several problems with the data and conclusions that are detailed below.

1) The authors show that GSCs from the mesenchymal (MES) subtype of glioblastoma (GBM) release greater numbers of extracellular vesicles (EVs) containing EGFR/EGFRvIII protein and mRNA as compared to the EV release by proneural (PN) GSCs, based on EV release from two MES GSCs and EV release from two PN GSCs (Fig. 1). The relevance of this finding to tumor aggressiveness or mouse survival is somewhat substantiated by the observation that xenograft tumors from one PN GSC have a much longer survival than xenograft tumors from two MES GSCs; however, the authors need to include the survival data from another xenograft tumor of the PN GSC subtype in this figure.

2) The authors evaluate angiogenesis and vasectasia in HUVECs (human umbilical vein endothelial cells) and mouse endothelial cells that have been treated with the EVs of MES GSCs and show apparent expression of EGFR and EGFRvIII in the endothelial cells (Fig. 2). The relative migration of EGFR/EGFRvIII is not clear in the western blots (EGFRvIII should have a relative migration of 145-kDa on reduced SDS PAGE and wild-type EGFR should migrate more slowly).

3) Studies with human brain microvascular endothelial cells should be included in Figs. 2 and 3, not just HUVECs.

4) In Fig. 4, the authors nicely show the enhanced survival effect of combined targeting of VEGF (with an antibody to VEGF) and an inhibitor of EGFR in the xenograft model of GBM, and the authors also show that this combined targeting appears to block formation of the large diameter blood vessels in the tumors. The authors need to show decreased phosphorylation of EGFR/EGFRvIII in these tumor vessels to convincingly demonstrate specific inhibition of pEGFR with the EGFR inhibitor.

5) The spatial transcriptomic findings from the 8 xenograft tumors (Fig. 5) where regions of interest (ROIs) showed differences in gene expression needs some validation of the key genes discussed in the manuscript as being relevant to the vascular processes focused on.

Reviewer #4 (Remarks to the Author):

The report by C. Spinelli et al elegantly demonstrated EGFR-dependent neovascularization of MES or PN-GBM subtypes. The main novelty of their research is the role of EVs containing EGFR mRNA, causing expression of tumor-associated EGFR on endothelial cells and subsequently pathological neovascularization. Based on the transcriptional subtype and potential occurrence of Chr 7 and EGFR amplifications, the tumors acquired distinct vascularization patterns, ranging from large vessels, namely vasectasia (EGFR-high), to very dense vascular networks (EGFR-low).

The paper is well written, and its hypotheses are clearly stated. The experiments are designed in a comprehensible way. In my opinion, the overall work is conclusive and addresses an important topic for the scientific community.

Despite the already mature impression of the work, I have some concerns and weaknesses that ought to be addressed in a revision.

1. the model and the transfer to humans

The main problem is the exclusive use of murine models (despite the use of human cell cultures). The authors might want to use some human data for validation or explore their hypothesis and results in a human setting. The use of cell type-specific transcriptomics is interesting and can be easily extended to human FFPE samples. What is the effect of BEV therapy in human samples? Or at least use some human datasets such as the latest single cell or spatial transcriptome data (for stRNA-seq, H&E could also help identify the different types/patterns of neovascularization).

2. cellular and microenvironmental heterogeneity.

The well-known cellular heterogeneity of GBMs is almost ignored in the presented work. I fully understand the use of preclassified MES/PN cell lines representing the majority of neural/oligo- or mesenchymal differentiated cell states for a model, but this limitation needs to be discussed in more detail. The different pattern of vascularization can also exist in the same tumor, suggesting that spatial heterogeneity or other spatially resolved confounding factors (such as metabolism or immunity) may influence or interact with the described results.

3. Genetic background and expression.

EGFR, together with Chr7 amplification, is the hallmark alteration of GBM. Cell lines often lose their initial high EGFR copy gain to a more moderate level. What is the level of EGFR gain in the cell lines studied? Is there a correlation between genetic (CNA) and EGFR (mRNA+) EVs? It would be interesting to study the vascular pattern of tumors with higher or lower EGFR Amp.

In conclusion, the work is of great interest. The translation to human tumors and various confounding factors based on tumor heterogeneity or microenvironment should be further explored and discussed.

December 18, 2023

Responses to Reviewers

We are grateful to all Reviewers for the time and insights that enabled us to make substantial revisions of our manuscript entitled: ***“Mesenchymal glioma stem cells trigger vascetasia, a distinct neovascularization process mediated by extracellular vesicles carrying EGFR”*** (Ms. NO. NCOMMS-22-17857A-Z) by Spinelli et al. This process is now completed to the best of our ability, and it involved extensive reworking of several parts of our paper, as well as numerous pieces of new data, as listed below. This work took us longer than we expected due to the extent of revisions we have undertaken to address the critique, as thoroughly as we could, but also because some of the authors experienced personal challenges during the intervening months. Still, we hope that the outcome will satisfy the Reviewers. Below is the detailed list of our revisions and responses to the Reviewers’ specific critique and a listing of revisions we have implemented.

The main changes introduced into the revised manuscript include:

Fig. 1: Major changes and new data in panels a, b, c and d

Fig. 3: Revised panel d

Fig. 4: Completely redesigned figure with new data including in panels a, b, g

Fig. 5: Redesigned figure

Supplementary Fig. 1: Completely revised with new data in panels e and f

Supplementary Fig. 2: Completely revised with new data in panels c, d and e

Supplementary Fig. 3: Revised

Supplementary Fig. 4: Revised panels a and b

Supplementary Fig. 5: Revised and redesigned.

Supplementary Fig. 6: Revised

Supplementary Fig. 7: Revised panels c and d

Supplementary Fig. 8: Revised

Supplementary Fig. 9: Completely redesigned panels d, e and f

Supplementary Fig. 10: Completely redesigned

Supplementary Fig. 11: New figure

Supplementary Fig. 12: New figure

Supplementary Fig. 13: New figure including new data on EGFR phosphorylation

Supplementary Fig. 14: New figure

Supplementary Fig. 15: New figure including new data on RNAScope analysis

Supplementary Fig. 16: New figure containing revised design of diagrams

Supplementary Fig. 17: New figure containing revised data items

Our point-by-point comments addressing the specific concerns of the Reviewers are detailed below.

REVIEWERS’ COMMENTS

Reviewer #1

This thorough and rigorous paper addresses mechanisms of angiogenesis in glioblastoma, contrasting proneural and mesenchymal angiogenesis. The paper elucidates potential reasons for failure of anti-VEGF glioblastoma therapy. It also provides compelling evidence from cell and PDX models for the role of extracellular RNA communication and Extracellular Vesicles (EVs) in cancer progression. Spatial single-cell profiling of PDX models integrates tissue-level and molecular perspective. Dual targeting of EGFR and VEGF points at specific combination therapies that may need to be applied in clinical trials. Overall, the paper is comprehensive, solid and of potentially high interest to Nature Communications audience. A number of minor issues and requests for clarification are listed below.

We would like to thank the Reviewer for this positive and thoughtful summation of our work. This is much appreciated.

1. Fig 1f– The sprouting for MES exceeds that of PN, which is not concordant with higher microvascular density of PN vs MES illustrated in 1d.

This is an excellent point. We believe that there is certain tension between semantics and biology of these assays that could be confusing, and we made the respective clarifications in the text and in figures.

In our hands, the aortic ring assay measures morphogenetic and growth responses of endothelial cells to various stimuli including growth factors (VEGF) and particulate mediators (EGFRvIII-EVs). These responses manifest themselves as outgrowths of endothelial structures that resemble angiogenic sprouts, but differ from them in several ways, such as the absence of remodelling or anastomosis, absence of blood flow and sheer force and lack of any obvious gradient of the angiogenic stimulus, which in this case is added exogenously to the entire well containing isolated aortic segments ¹. Since angiogenic gradient normally drives the directional migration of endothelial tip cells *in vivo*, a prerequisite of bona fide sprout formation ², in the absence of such spatial cue the aortic ring assay largely measures endothelial cell growth/migratory activation, in somewhat analogous way as do other surrogate assays, such as transwell endothelial cell migration (used in our study), or *in vitro* tube formation (used by others ³). In none of these instances endothelial cells form functional sprouts that result in development of complete capillaries, an end result of angiogenesis. To avoid confusing analogies between these surrogates and blood vessel formation processes *in vivo*, we decided to use the names of “aortic ring assay” and “endothelial outgrowths”, rather than “sprouting assay” and “endothelial sprouts”.

In contrast to the aforementioned measurements of endothelial *ex vivo* responses, Fig. 1b-d depicts vascular patterns present *in vivo*. Naturally, this snapshot captures the entirety of the tumour blood vessel dynamics, including structures that have already become functional, as well as those that are dynamically emerging. While the surrogate ‘angiogenesis’ *in vitro* assays, as mentioned earlier, may capture some of the underlying biology (endothelial responses), and could be useful for analytical purposes, they do not

reflect the entirety of processes that lead to formation of vascular networks, especially the enlarged vessels (*vasectasia*). We are in the process of developing a three-dimensional assay that may enable modelling such non-angiogenic vascular responses, but this is not a trivial undertaking, and the work is still ongoing.

We believe that one important aspect that our study brings into this realm is the realization that not all blood vessel formation processes *in vivo* are tantamount to angiogenesis resulting in formation of dense network of capillaries⁴. In fact, in the case of tumours formed by proneural glioma stem cells (PN-GSCs), in the secretome of which vascular endothelial growth factor (VEGF) is a dominant activity (Fig. 1k), such capillary networks can be visualised morphologically (Fig. 1b-d) and documented molecularly by expression of tip cell markers such as Apelin and VEGFR2 (Fig. 4g).

In contrast, as we report in the present manuscript, endothelial stimulating activity in the secretome of mesenchymal glioma stem cells (MES-GSCs) is profoundly different, and results in formation of enlarged and sparse blood vessels through a non-angiogenic process we termed '*vasectasia*'. This process is driven, at least in part, by extracellular vesicle (EV)-mediated transfer of oncogenic EGFR and entails circumferential growth of vascular structures in the absence of tip cell markers. Instead, *vasectasia* is marked by other molecular traits, such as the expression of Birc5 and other hallmarks (new Fig. 4g; Supplementary Fig. 15a-b). As endothelial cells proliferate during both, angiogenesis and *vasectasia* (new Fig. 4g), we believe that aortic ring assay does not distinguish between these two processes. We made these considerations clearer in the revised version of our manuscript.

2. Fig 1i and j, also Supp F1c, suggest MES EVs stimulate VEGF-dependent sprouting without VEGF protein being detectable within them. This should be addressed. Are there mRNAs, miRNAs or other noncoding RNAs in these EVs that may explain this activity?

We apologize for the lack of clarity in the design and description of these figures. Those have been corrected in the revised submission. In fact, the endothelial growth responses (aortic ring assay) shown in Fig. 1f and 1g are of global nature, regardless of the presence or absence of VEGF in the secretome of the respective cancer cells. We did not intend to suggest that they are VEGF-dependent. More specifically, Fig. 1j shows that EV preparations of PN-GSCs contain little endothelial stimulating activity, but they do express such an activity in the EV-free supernatant (Fig. 1i). This is consistent with the observation that supernatants of these cells contain appreciable quantities of VEGF, which were, in fact, equal or somewhat higher than those released by MES-GSC (Fig. 1k). In all those assays, recombinant VEGF (25 ng/mL) was used merely as a positive control. Thus, all glioma stem cell lines produce soluble VEGF, but this factor is (in our hands) essentially undetectable in all tumour EVs that we have analysed (Fig. 1j). In spite of the absence of VEGF in them, EVs from MES-GSCs stimulate endothelial growth and migration, while EVs from PN-GSCs are unable to do so.

These observations are consistent with the notion that MES-GSC-derived EVs contain a VEGF-unrelated blood vessel stimulating activity that parallels *vasectasia* (and not angiogenesis) *in vivo* and (unlike PN-GSC EVs) triggers phosphorylation of EGFR in

cultured endothelial cells. These observations led us to studies on the link between EV-associated oncogenic EGFR (mRNA) and the new GBM-driven vascular process (*vasectasia*) that we described in the present study. We hope that these clarifications and the corresponding alterations in the text and figures will make these connections easier to appreciate.

3. Fig2b – What is the positive control for the effect of PN EVs on endothelial cells?

We realise that the absence of EGFR transfer to endothelial cells treated with PN-GSC EVs, as depicted in **Fig. 2b** may seem puzzling due to the absence of EV-related signal. This is largely dependent on the nature of PN-GSC EVs, which are not only devoid of EGFR (as are their parental cells), but also do not carry common exosomal markers, as we have observed and published previously ⁵. However, in the same study, we described a transfer of EV-associated fluorescence by PN-GSC EVs (on par with MES-GSC EVs ⁵). We made a reference to this observation in the current manuscript.

Since in the present study we have not observed appreciable biological effects of PN-GSC-derived EVs on endothelial cells we did not invest in seeking which of their constituent proteins or nucleic acids are efficiently transferred to recipient cells. However, we did monitor the experiments shown in **Fig. 2b** through the loading control of EV recipient cells, for which we used beta actin (B-actin) and by comparison (in the same gel) between endothelial cells treated with equal amounts of EVs derived from either PN-GSCs or MES-GSCs (or untreated).

Moreover, to further document the EV-mediated protein transfer from PN-GSCs to endothelial cells, we focused on CD133, which is a marker of PN cells. Thus, we incubated human PN-GSC EVs with mouse endothelial cells (EOMA) and probed them for human specific CD133. As expected, the human CD133 protein was readily detectable in GSC EVs and EV-treated mouse EOMA cells (**Supplementary Fig. 2d, left panel**) suggesting intercellular transmission. The transfer and uptake of PN-GSC EVs was also validated by fluorescent labelling of these EVs. Thus, we used APC-CFSE to label PN-GSC-EVs before incubation with mouse endothelial cells and measured the fluorescence transfer by flow cytometry (**Supplementary Fig. 2d; right panel**).

We believe that these experiments illustrate the notion that only MES-GSC EVs transfer the EGFR/EGFRvIII expression from cancer cells to endothelium, while PN-GSC-EVs transmit other molecules albeit without a major effect on biological responses on endothelial cells. Hopefully our related clarifications will be found convincing.

4. Fig3c – VEGF is shown but legend says EGFR only. Moreover, the figure shows that MES EVs promotes sprouting and not vasectasia, as would be expected from discussions elsewhere in the paper.

Once again, we must apologize for our lack of clarity. What we intended to show in **Fig. 3c**, is that endothelial cells (in aortic rings) were stimulated with EVs from either wild-type MES-GSCs (GSC83, GSC1005), or from their counterparts with disrupted EGFR (KO). In

this setting, the key variable is the content of EGFR/EGFRvIII in EVs, while VEGF is added as a positive control to merely show that the assay has worked. VEGF is used in this regard as a known endothelial stimulator, expected to elicit growth responses of endothelial cells. We have now revised the text, legends and figures to remove these troubling ambiguities.

As we commented in earlier paragraphs, in our hands, the aortic ring assay measures the growth/migration capacity of endothelial cells and does distinguish the ability of various stimulants to drive different forms of vascular response. For example, this assay (like many others) does not capture the various morphogenetic events that separate angiogenesis from other blood vessel growth processes, including *vasectasia*. The latter can be visualised *in vivo*, either in tumour tissue (e.g. Fig. 1b-d), or in Matrigel/BME implants containing MES-GSC-derived EVs that carry oncogenic EGFR/EGFRvIII (Fig. 3e; Supplementary Fig. 6).

At the present time, to the best of our knowledge, there is no *ex vivo* assay for detection and measurement of *vasectasia* (our present paper describes this process for the very first time). Our new data included in the present set of revisions, document that *vasectasia* entails circumferential expansion/proliferation of vascular structures engulfed by tumour parenchyma expressing oncogenic EGFR (new Fig. 4a-b). It is possible that once molecular and cellular features of *vasectasia* are more completely understood, microfluidic or organ-on-chip models could be developed to reflect this process more accurately (our explorations in this regard are ongoing). We inserted some of these clarifying comments into the new version of our manuscript and hope that they facilitate clearer perception of our findings.

5. Fig3d – *In light of emphasis of the paper on EGFR effect on vasectasia, in contrast to angiogenesis, why was vasectasia not assayed? Why would the effect of EGFR on migration be more related to vasectasia than angiogenesis?*

These are very fair questions. We used *in vitro* assays of endothelial activity/function (commonly known as ‘angiogenesis assays’, such as aortic ring, or transwell migration) mainly for reductionist purposes, for example, to demonstrate EGFR-associated ‘activity’ of GSC-EVs. Based on our observations, these assays are unable to distinguish between very different programs triggered in endothelial cells by fundamentally different mediators, such as VEGF and EGFR-EVs. We would like to suggest that, in spite of their name (‘angiogenesis assays’), these assays measure a combination of basic endothelial cell proliferation and migration responses, which are probably common for many vascular processes and likely necessary, but insufficient, to explain formation of functional blood vessel structures, be it through angiogenesis, or other processes (arteriogenesis, intussusception, arteriogenesis, remodelling, or cooption) including *vasectasia*. Arguably, the canonical pathway of angiogenesis, as it is known today (VEGF/VEGFR2, Angiopoietins/Tie2) would have not been discovered on the basis of these assays alone, and required a combination of *in vivo* and *in vitro* studies⁴. Therefore, as Reviewer rightfully suggests, once the molecular biology of *vasectasia* is more consolidated, designing proper and informative assays for this form of vascular growth would be a high priority for future studies.

At the present time, we have traced *vasectasia* through two types of assays: (i) analysis of vascular patterns in cranial xenografts of MES-GSCs and (ii) analysis of vascular structures in subcutaneous Matrigel/BME plugs containing EGFR-EVs. Using these read outs, we observed that while EGFR-EVs stimulate migration and proliferation of endothelial cells *in vitro* (aortic ring and transwell assays), they do not trigger angiogenesis on the basis of the following criteria: (i) Addition of EGFR-EVs does not activate canonical angiogenic pathways (VEGFR2, Tie2, PDGFRb) in cultured endothelial cells; (ii) Depletion of EGFR from MES-GSC EVs (CRISPR/Cas9) leads to a disappearance of enlarged vessels in brain tumours, but without affecting capillaries; (iii) Addition of EGFR-EVs to Matrigel plugs triggers *vasectasia* (in the absence of cancer cells), while similar plugs containing soluble VEGF exhibit capillary blood vessel pattern indicative of angiogenesis; (iv) In Matrigel (BME) assays we were able to visualise human EGFR in endothelial cells of large vessels recruited into the plugs; (v) Inhibition of EGFR kinase activity (Dacomitinib) in MES-GSC xenografts (*in vivo*) results in selective elimination of large vessels, but did not affect small angiogenic vessels, while treatment with VEGFR2 inhibitor (DC101) had the opposite effect; (vi) According to newly generated data we observed profound remodelling, enlargement and endothelial proliferation of individual cerebral vessels, as they enter tumour mass at the boundary of normal brain parenchyma and expanding glioblastoma xenografts expressing EGFR/EGFRvIII. Indeed, around Day 9 of tumour formation, the predominant features of blood vessels include their circumferential extension rather than angiogenic branching of new capillaries (new Fig. 4a-b). This results in low density of enlarged vessels in MES-GSC xenografts (Fig. 1b-d); (vii) Also new data related to both spatial RNA sequencing and single cell RNA sequencing of tumour (MES-GSC)-associated endothelial cells revealed that small, angiogenic blood vessel endothelial cells are enriched in Apelin and VEGFR2 (tip cell markers), while large *vasectasia*-related tumour blood vessels are enriched in survivin (Birc5) along with a set of other genes.

We realise that our use of what has become known as ‘angiogenesis’ *in vitro* assays may seem ambiguous, but while we do not have yet an *in vitro* assay that would accurately model *vasectasia*, we interpret the assays we used as merely the ability to stimulate basic endothelial cell responses. With these additional clarifications and interpretation, we trust that the Reviewer will find our findings convincing.

6. scRNA-seq of EGFR+ and EGFR- xenografts reveals distinct EGFR+ and EGFR- clusters with no cluster spanning EGFR+ and EGFR-. This pattern is highly conspicuous and may possibly be explained by a batch effect due to separate handling and sequencing of EGFR+ and EGFR- tumors. For the current explanation to stand, such batch effect needs to be ruled out.

This is a valid point, which we have given extensive consideration. We have revisited our datasets and validated some of the resulting differentials at the protein level, including staining of relevant tissues for angiogenic tip cell markers (Apelin, VEGFR2) and emerging *vasectasia* markers that were revealed by single cell RNA seq experiments (new Supplementary Fig 11).

In addition, we reviewed the data with bioinformaticians and co-authors (Minjun Kim and Yasser Riazalhosseini) and revised the manuscript by highlighting measures to eliminate

misinterpretation of single cell RNA seq data, including batch effects. Thus, all experimental procedures except the sample collection, which was inevitable due to the different clinical endpoints, were carried out together from cell injection to sequencing. Even though there is a possibility of technical biases by a different batch status, our results suggest that the distinct separation between EGFR+ and EGFR- clusters is associated with their experimentally demonstrable biological differences (tumour growth, vascular patterns; Fig. 3f-i), rather than a batch effect.

In our study, as much as possible, we also followed the standards in the field. For example, the recent study by Aissa et al ⁶ showed transcriptomic changes associated with response and tolerance to the EGFR inhibitor, Erlotinib, in lung cancer using single-cell RNA-seq. In this case, multiple cell lines treated with Erlotinib exhibited clearly distinct clusters depending on the state of EGFR inhibition. This is not unexpected, also in our case, since EGFR has strong downstream effects on the overall transcriptome and cellular/tumour phenotype. Interestingly, genes upregulated by EGFR inhibition in *Figure 2e*, of the paper by Aissa (CALD1, CCDC80, TPM1, and IGFBP3), are also consistently upregulated in our datasets for EGFR-KO MES-GSCs. This would suggest that the distinct pattern of UMAP clusters, as a function of the EGFR status, is clearly derived from biological differences, rather than a batch effect. We have added the respective comments to the revised manuscript.

7. 5e,g: It is not clear how 83 WT and 83 KO add up to 100%. How are the percentages specifically calculated?

Thank you for drawing our attention to this aspect. In these experiments, clustering analysis was performed on endothelial cell populations from both EGFR-WT and EGFR-KO tumors, and each cluster was labeled based on their distinguished marker genes over the others. Thus, each cluster in Fig 5e contains endothelial cells from both groups adding up to 100% to visualize the composition of distinct subsets of endothelial cells with different functional identities to the overall pool. This proportion was presented as a function of the EGFR status of adjacent cancer cells. Again, we vetted this approach with bioinformaticians and hope this was of presenting the data is acceptable.

8. Lines 269-270: The sentence may be mis-interpreted to mean that angiogenesis is just “vessel enlargement”. Why is vascetasia not part (or an aspect of) angiogenesis?

We apologize for this confusion. We clarified the sentence in the text and added explanatory comments to the extent the available space permits.

The question as to the conceptual content of the widely used term “angiogenesis” and whether it could accommodate ‘vasectasia’ is important, and we would like to address it even briefly. First, according to presently accepted definition, angiogenesis is viewed as formation of new capillary which branches from pre-existing vessels largely through either formation of tip cell containing vascular sprouts, or alternatively by vessel splitting (intussusception). The canonical pathway driving the sprouting angiogenesis program of

endothelial cells relies on VEGF/VEGFR2 axis, which propels organized and directional endothelial cell growth, migration and morphogenesis ⁷.

Our data suggest that while *vasectasia* also entails vascular growth, this process does not lead to vessel branching, or elongation, but instead entails circumferential enlargement of the existing thin-walled capillary vessels upon entry into the tumour microenvironment from the surrounding brain parenchyma (new Fig. 4a-b). Since no branching, or splitting occurs, or no tip cell phenotypes (e.g. VEGFR2+) emerge, and targeting VEGF has no effect in this case, this process does not fall into the conceptual framework of angiogenesis, and it is, actually, quite unique, as is its inducing factor, EGFR-EVs.

It could be argued that compressing many biological meanings into few terms, such as it occurred around “angiogenesis”, resulted in influential and wide-spread perceptions that, one might say, impeded research and medical progress. For example, the assumption that all vascular processes in cancer can be reduced to angiogenesis along with the discovery of the canonical VEGF pathway driving angiogenesis, resulted in partially failed attempts to use VEGF inhibitors as therapeutics across all cancers. While solid tumours are almost invariably vascularized, this important property, arguably, could be acquired through several distinct (possibly targetable) mechanisms, of which we believe *vasectasia* is one. While it would be difficult to elaborate on this at length in the manuscript, we strove to be very clear in that regard, and are grateful that Reviewer has raised this point to help us make this more transparent.

We are also indebted for all other insights and helpful comments that led to considerable improvements in our manuscript.

Reviewer #2

Mesenchymal glioma stem cells trigger vasectasia, a distinct neovascularization process mediated by extracellular vesicles carrying EGFR, C. Spinelli, L. Adnani, B. Meehan, L. Montermini, S. Huang, M. Kim, T. Nishimura, S.E. Croul; Y. Riazalhosseini and J. Rak

Glioblastoma multiforme is the most deadly form of malignant tumor rendering a median patient survival period of under 15 months despite undertaking currently available treatments. Transcriptional profiling has allowed the classification of GBM tumors into four subtypes: classical, mesenchymal, proneural and neural, stemming from glioma stem cells (GSCs), the GBM tumor initiating cells divided into proneural (PN) and mesenchymal (MES) subtypes, contributing to the heterogeneity in GBM. The differences in transcriptional expression of the epidermal growth factor receptor (EGFR) observed in GBM has been linked to the aggressiveness of GBM tumors presumably due to its role in cell proliferation, migration, angiogenesis and survival. Therefore, a better understanding of the role of EGFR in GBM would be beneficial towards the development of GBM treatments.

In the manuscript under review, Rak et al demonstrated that in comparison to PN-GSC xenografts, the stunted survival of tumor bearing mice with MES-GSC xenografts was related to the differences in vascular patterning resulting from EGFR expression in MES-GSC extracellular vesicles (EVs). The presence of pEGFR specific to the MES-GSCs was deemed responsible for the formation of a

less dense vasculature with larger vessel formation as observed through fluorescence imaging of Anti-CD31 in endothelial cells. Since mouse aortic rings stimulated with culture media from GSC cultures and with pelleted EVs (isolated through centrifugation of corresponding culture media) demonstrated an increased sprouting response in both cases, further experiments were performed to decipher what factors are responsible for the spout induction in both circumstances. One such factor was discovered to be VEGF, but its expression remained exclusive to culture media and absent in EVs. Therefore, further experiments were conducted to assess what factors in EVs are responsible for the development of this effect. Therein, the phosphorylation of EGFR was detected in endothelial cells stimulated with EVs from MES-GSCs and Western Blot analysis demonstrated that EGFR (and the GBM-specific mutant EGFRvIII) protein and mRNA expression may be transferrable from EVs to endothelial cells. Negative controls such as the EGFR inhibitor Dacomitinib and EGFR/EGFRvIII gene knock-out in MES-GSCs resulted in a decrease in endothelial cell migration, sprouting, increased the survival of tumor bearing mice and reverted vasculature changes thereby proving the importance of EGFR expression in MES-GSC EVs responsible for the observed tumorigenic enhancement in this GSC subtype. Additionally, single cell transcriptional profiling was conducted for MES-GSC xenografts positive and negative for EGFR and these results classified endothelial cells into migrating, proliferative, permeable and angiogenic subpopulations. Finally, the top 100 genes variably expressed between the large and small vessels of EGFR-WT tumors, EGFR-KO tumors as well as control brain tissue were listed. In conclusion, the authors described the existence of two different GSC-mediated blood vessel growth patterns: PN-GSCs that lack the presence of EGFR producing dense capillary networks through the mediation of angiogenesis and MES-GSCs that carry EGFR/EGFRvIII in EVs triggering ‘vasectasia’, a term they coined for the observed circumferential extension of tumor blood vessels. Correspondingly, the authors proposed a novel approach to GBM therapy by combinatorial targeting of VEGF/VEGFR2 and the EGFR pathway through the administration of the inhibitor DC101 and inhibitor Dacomitinib, respectively, and demonstrated its therapeutic potential by presenting a prolonged lifespan of MES-GSC xenograft tumor bearing mice simultaneously reverting their vasculature changes to the PN-GSC phenotype. I believe that this manuscript is promising and the experiments are very well conducted. However, I have written a few suggestions below that would be beneficial to strengthen the premise in this paper.

We very much appreciate this thorough, accurate and positive summation of our work by the Reviewer. This synopsis is excellent also because it highlights the notion we intended to suggest in our paper, namely that, in spite of certain popularity of VEGF-driven tumour angiogenesis in the literature, it is important to discover alternative processes of blood vessel formation in cancer and their alternative unique mediators. Among them, we suggest there are oncogenic EGFR-carrying EVs which may be involved in the process that we refer to as ‘vasectasia’. We are very thankful for commenting on this.

Suggestions for improvement:

1. A study by Reardon et al from 2010 illustrated the outcomes of VEGF and EGFR combinatorial targeting in GBM patient clinical trials utilizing the administration of Bevacizumab with Erlotinib5. The outcome prognosis was not ameliorated in comparison to historical Bevacizumab consisting treatments. I recommend elaborating why Dacomitinib and DC101 treatment would be superior for consideration in GBM treatment in comparison to previously attempted combinatorial drug approaches.

This is an excellent question, which absolutely requires a commentary here and in the revised text. We believe that there are several reasons why a combination of EGFR

inhibitors and inhibitors of the canonical angiogenic pathway (VEGF) may deserve a second look, in part based on our data.

There are at least three major factors that fundamentally separate our xenograft experiments from the context of the pivotal clinical trial conducted by the Reardon group⁸ (other than our study being experimental). First, at that time of these trials (and also thereafter), GBM patients were not stratified according to their molecular (or glioma initiating cell) subtype or, more importantly, according to predominant vascular processes underlying disease progression. Second, these studies, largely for ethical reasons, were conducted with patients with recurrent disease, where mutational landscapes and cellular phenotypes would have been profoundly impacted by preceding radiation, temozolomide chemotherapy, and disease progression, while the related vascular driving secretomes and vascular patterns would have undergone untold number of unknown and hitherto largely unstudied changes.

The third factor is the drugs that were used. Erlotinib is the first-generation, reversible EGFR inhibitor with limited activity against the EGFR pathway in GBM settings⁹. In contrast, Dacomitinib, has shown an encouraging, albeit limited, signal in a subset of GBM patients^{10,11}. This is an oral highly selective quinazoline, and the second-generation, irreversible EGFR tyrosine kinase inhibitor that to our knowledge has not been tested clinically in GBM patients in combination with VEGF antagonists, especially those with treatment naïve and molecularly defined disease. Therefore, while the principle of EGFR targeting may have been similar the means of doing so was very different between our experiments and these clinical studies.

While we have limited space to further elaborate on several fascinating aspects of these EGFR targeting agents in GBM, and in MES-GSCs in particular, it could be added here that the half-life of Erlotinib is also significantly lower than that of Dacomitinib¹², and that Erlotinib could stimulate nuclear receptor subfamily 1 group I member 2 (NR2F1), which could promote the metabolism of the drug itself and may potentially have implications in the growth and/or chemo-resistance of cancers¹³.

The addition of an antiangiogenic agent in our experimental protocol may also require one few additional words of clarification. In the aforementioned pivotal trial conducted by the Reardon group⁸, the agent of choice was Bevacizumab. This was likely predicated on the, then accepted, notion that VEGF driven angiogenesis is the dominant (if not the only) relevant neovascularization process with implicit significance in the highly vascular context of GBM^{14,15}. As mentioned earlier, the subsequent phase III studies with treatment naïve GBM patients^{16,17} put this concept into question, without revealing the actual nature of vascular growth in GBM, or alternative modes of its targeting. Of note, while Bevacizumab did not prolong overall survival of GBM patients, it did change their vascular patterns by eliminating small (angiogenic) vessels and leaving larger vascular structures intact^{16,18}, the latter observation consistent with our results. Whether these larger vascular structures emerge as a consequence of '*vasectasia*', remains to be elucidated potentially through the use of markers we described (e.g. survivin).

Furthermore, these pivotal trials contain multiple complex elements that our experiments were not meant or designed to capture. Instead, we undertook a targeting approach focusing on the canonical VEGF angiogenesis pathway, as one well-studied ‘control’ to our exploration of the new vascular growth mechanisms involving ‘*vasectasia*’. Our inclusion of DC101 antibody, which targets mouse VEGFR2, was predicated on three factors: (i) the canonical role of VEGF/VEGFR2 pathway in the brain vasculature ¹⁴; (ii) the appreciable content of VEGF in the conditioned media of all GSCs we analysed (but not in EVs); (iii) the preponderance of capillary, angiogenic-type small vessels in PN-GSC driven xenografts, along with their markedly lesser presence in MES-GSC xenografts in mice, unless EGFR was disrupted or antagonized. We realise that DC101 is not fully analogous to Bevacizumab from the 2010 trial, and it could resemble the subsequently introduced Ramucirumab (anti-human VEGFR2 antibody). However, we felt that DC101 represents an optimal experimental paradigm, as it would block both cancer and stromal derived VEGF from binding to its crucial angiogenic receptor (VEGFR2).

All this is to say that our experiments were meant to illustrate a biological paradigm of co-targeting two different vascular processes (angiogenesis and *vasectasia*) in GBM models and they, to our knowledge, have no direct precedent in the clinical trial literature. We hope this is more clearly articulated in our revised manuscript.

2. J. Rak has previously published reports describing that GBM cells release EVs and that glioma cells release membrane-derived microvesicles consisting of oncogenic EGFRvIII (6,7). I recommend elaborating why studying the phenomenon of EGFR presence in MES-GSC EVs is superior or significant for understanding GBM therapeutic approaches in comparison to already known effects of EGFR derived from microvesicles in GBM cells.

This is another extremely important point, for raising of which, we are truly grateful. The short answer to Reviewer’s thoughtful query is that the current study demonstrates a completely (structurally and molecularly) novel, cancer-specific and EGFR-EV-driven vascular growth process (‘*vasectasia*’). This was not the substance of prior reports.

Moreover, in our hands, *vasectasia* entails the transfer of EGFRvIII mRNA to recipient endothelial cells, followed by the expression of exogenous phosphorylated EGFRvIII and changes in expression of multiple downstream genes (e.g. survivin) along with non-angiogenic morphological responses. Prior studies simply reported the EGFR protein content in cancer EVs.

We suggest that *vasectesia* is unique to a specific subtype of glioma stem cells, notably those with mesenchymal signature, cells that may be present during, or dominate, the GBM progression. This has not been appreciated previously to our knowledge. In this sense ‘grafting’ of oncogenic EGFRvIII mRNA and the resulting expression of the related, constitutively active protein, amidst the signalling apparatus of endothelial cells may abort their angiogenic program and produce cancer-specific morphogenetic and molecular responses, which were not previously observed or described.

The present study also provides a path to recognizing and therapeutic translation of these findings. In a more general sense, our study proposes that in complex cancers, such as GBM, multiple vascular growth processes may drive disease biology in ways that are not reducible to “angiogenesis”, and even less to canonical VEGF responses. Moreover, our study suggests that rather than assuming that blood vessel targeting in GBM “does not work” (as one often hears) it may be worth reflecting on a deeper biology and multiplicity of modifiable pathways, through which the pivotal vascular component is affected in brain tumours.

This perspective, we believe, represents a quantum leap from what we and others have published thus far on various aspects of EV-driven “angiogenesis”. Indeed, we have earlier proposed that EGFR/EGFRvIII oncoproteins can be included into, and transferred between cells as, cargo of extracellular vesicles (EVs). We suggested using established carcinoma cell lines that this process may impact VEGF production and other features of surrounding cells, including endothelial cells ^{19,20}. This is not what we find using patient derived GSC isolates.

Our earlier studies make no reference to *vasectasia*, its molecular features or role in GBM progression. We also ‘missed’ the intercellular transfer of EGFRvIII mRNA in our prior work. This transmission from glioma stem cells and the profound impact this transfer has on endothelial cells were never described to our knowledge, again, to say nothing about ‘*vasectasia*’ *in vivo*. We did not know such a process exists.

Moreover, we (and others) made no connection between molecular subtypes of GBM lesions, and between glioma stem cells and their driven differential vasculature modifying pathways. Neither has impact of a mutant oncogene on the heterogeneity of endothelial cell subpopulations associated with angiogenesis *vs* vasectasia been described or mentioned, to the best of our knowledge. Finally, the present study sets a paradigm for other processes where oncogenes may be transferred to blood vessel and stromal cells and impact the architecture of the tumour microenvironment in a matter dependent on molecular ‘wiring’ of the underlying cancer.

We believe that just as there are multiple studies on various effects of VEGF in tumour angiogenesis (over 4,000 citations on PubMed), the role of EGFR/EGFRvIII and other oncogenic pathways in regulating the tumour vasculature is an evolving field, in which our study culminating in the discovery of EGFR-EV-driven *vasectasia* represents, we believe, a significant leap forward. We have highlighted this in the revised version of our manuscript and hope the Reviewer is convinced that we bring a new perspective into the realm of brain tumour studies.

Minor suggestions:

1. More attention should be directed towards the use of proper units. E.g., micro should be written as ‘μ’ instead of ‘u’ as observed in lines 172, 318, 333, 376, 379 and Figure 4b.

Our apologies for these errors, which have now been corrected.

2. *In Figure 2, although the images demonstrate Western Blot results, there is no indication in the text as to how many times these experiments were repeated ('n').*

The number of repeats has now been added to figure legends. Thank you for pointing this out to us.

3. *Certain figure y-axis labels should include units of measurement or these units should be described in the text (e.g. Figure 4d).*

Once again, this is our unfortunate omission, which has now been corrected. Thank you.

Discretions:

1. *Certain figures would benefit visually from improved symmetry and alignment of statistic indicators or axis labels (e.g. Figure 4b, d).*

This is a great suggestion; much appreciated. We have redesigned multiple figures to accommodate new data, and in the process, we made every effort to make the visuals more harmonious and symmetrical, to the best of our ability.

2. *In certain experiments that portray a reduced specific effect after knocking-out the EGFR gene or use of inhibitor Dacomitinib (Figures 3b-d) it would be beneficial to demonstrate the statistics indicating the diminution in effect after obliterating EGFR function by comparing the results to the WT control rather than just comparing the values to the untreated control.*

Thank you for this very useful comment. We addressed this problem by introducing values for single replicates in addition to the overall statistic, which we hope improves the reception of these results.

3. *In the description for Figure 5, various upregulated genes are listed. It would be beneficial to the reader if the significance of these genes was briefly described. What are the roles of these genes? Are they in any way related to one another?*

We completely agree with this comment. While we need to respect space restrictions, a better description of some of the genes has been added to the text, as much as possible.

In this regard we strove to be concise and not overly speculative. Thus, we made brief comments throughout the text as to the genes with known vascular function, such as VEGFR2 and Apelin, which are markers of tip cells and play a role in canonical sprouting angiogenesis⁷. We were somewhat more cautious about the genes that were selectively upregulated in endothelial cells undergoing vasectasia, including Birc5, Socs2 and Srsf2. This is because, at this time, there is little mechanistic information as to how these newly identified genes and proteins contribute to vascular enlargement. We did not elaborate on their possible involvement in the manuscript to conserve space and to avoid an overly

speculative language, but we would like to say a few words in paragraphs below to briefly share with the Reviewer our initial thoughts.

Perhaps the most intriguing among these genes is Birc5, also known as survivin. This protein is a member of the family known as inhibitors of apoptosis (IEPs) and was found to play a role in endothelial cell survival during earlier studies, in which we were involved²¹. Survivin also possesses different regulatory roles, as a part of Akt, PDGFR and other signalling pathways in cancer cells and beyond²². Intriguingly, survivin is a regulatory target of EGFR²³. We would like to speculate that upregulation of survivin may represent the effect of ectopic expression of EGFR in endothelial cells, which may be part of the mechanism that, we think, aborts angiogenesis and drives endothelial cell survival required for *vasectasia*. Of course, this is presently unproven, and related studies are ongoing.

Socs2 is also interesting. This gene belongs to suppressors of cytokine signalling, including the JAK/STAT pathway and other effectors, with known roles in immunomodulation²⁴. Socs2 also binds to growth factor receptors, including EGFR and may be involved in mutual regulation²⁵, but the role of this mechanism in endothelial cell function is poorly studied. Notably, a member of this family, Socs3, was reported to inhibit angiogenesis²⁶, so it could be speculated that, in the course of *vasectasia*, Socs2 might contribute to switching from growth factor-driven sprouting growth to circumferential expansion pattern. Future studies may establish whether Socs2 is a major effector, or merely a marker of endothelial cells induced to undergo *vasectasia*.

Srsf2 (Serine And Arginine Rich Splicing Factor 2) gene product is a part of the spliceosome, and is known to regulate pre-mRNA splicing and mRNA stability²⁷. The role of SrsF2 in endothelial biology and angiogenesis is not very well established with emerging studies that suggest the involvement of this factor in splicing and expression of VEGFR1²⁸, and alternative splicing of VEGF165 to VEGFR165a²⁹. Both alternatively spliced VEGFR1 and VEGF165a possess antiangiogenic activities and therefore one may speculate that if Srsf2 affects these processes in the brain vasculature this may be a part of endothelial response to a switch from angiogenesis to *vasectasia*.

These suggestions are presently hypothetical, as are the clues emerging from spatial sequencing and other approaches, we have undertaken to understand *vasectasia* more fully. We are pursuing these clues, but we hesitate to elaborate on several genes of interest more extensively in the manuscript mainly due to lack of space and lack of solid data to back up our intuitions. We hope the Reviewer would find this approach acceptable.

Reviewer #3:

The manuscript by Spinelli et al focuses on angiogenesis versus vasectasia, the latter suggested to be induced by extracellular vesicles (EVs) from mesenchymal (MES) glioma stem cells (GSCs), and the role of EGFR/EGFRvIII in promoting this distinct vascular phenotype called vasectasia. Overall the manuscript is very interesting and contains novel

data that are of physiologic relevance. However, there are several problems with the data and conclusions that are detailed below.

We are thankful for this generally positive reception of our manuscript and findings, and we do appreciate the thorough analysis of the content and suggesting ways in which our work could be improved. We followed these recommendations and included several new pieces of data, clarifications and corrections as detailed below.

1) The authors show that GSCs from the mesenchymal (MES) subtype of glioblastoma (GBM) release greater numbers of extracellular vesicles (EVs) containing EGFR/EGFRvIII protein and mRNA as compared to the EV release by proneural (PN) GSCs, based on EV release from two MES GSCs and EV release from two PN GSCs (Fig. 1). The relevance of this finding to tumor aggressiveness or mouse survival is somewhat substantiated by the observation that xenograft tumors from one PN GSC have a much longer survival than xenograft tumors from two MES GSCs; however, the authors need to include the survival data from another xenograft tumor of the PN GSC subtype in this figure.

We completely agree with this recommendation, and we have included new survival data of the second PN GSC line (GSC528) in **Fig. 1a**. Both PN-GSC xenografts (GSC157 and GSC528) result in formation of intracranial tumours and glioma-like disease that is significantly less aggressive, as measured by survival time, than their MES-GSC counterparts (GSC83 and GSC1005). In the case of newly included GSC528 model, the survival time is approximately 82 days post injection versus under 30 days for MES-GSCs.

It should be mentioned that the patient derived GSC lines we used throughout this study were profiled and assigned the molecular subtype independently of the GBM tumours from which they were isolated ³⁰. We made this clearer in the revised version of the manuscript.

2) The authors evaluate angiogenesis and vasectasia in HUVECs (human umbilical vein endothelial cells) and mouse endothelial cells that have been treated with the EVs of MES GSCs and show apparent expression of EGFR and EGFRvIII in the endothelial cells (Fig. 2). The relative migration of EGFR/EGFRvIII is not clear in the western blots (EGFRvIII should have a relative migration of 145-kDa on reduced SDS PAGE and wild-type EGFR should migrate more slowly).

This is a fair point which resulted from our poor labelling of the respective Western Blot gels. This is now fixed. Specifically, in the revised manuscript, we provided a more accurate molecular weight (mobility) labels that enable highly reproducible identification of the corresponding wild type EGFR (slow migrating) and EGFRvIII (fast migrating) isoforms of this receptor.

It may be worth remarking that our data suggests both, EGFR and EGFRvIII transcripts are detectable in MES-GSC-derived EVs and both corresponding proteins are subsequently detectable in EV-treated endothelial cells, resulting in functional changes in these cells that we document to be linked to the process we refer to as ‘*vasectasia*’.

3) *Studies with human brain microvascular endothelial cells should be included in Figs. 2 and 3, not just HUVECs.*

We very much appreciate this thoughtful suggestion. Studies on the transfer of EGFR/EGFRvIII into brain derived microvascular endothelial cells exposed to MES-GSC-derived EVs have been added to Supplementary Figs. 1f and 2c-e. We would also like to mention that according to our observations thus far, *vasectasia* is driven by EGFR-EVs not only in the brain tumour vasculature, but also at other anatomic locations (it is EGFR-EV-specific, but not brain-specific). For example, addition of EGFR-EVs to BME (Matrigel) implants (plugs) injected subcutaneously, also lead to formation of enlarged blood vessels upon recruitment of cutaneous endothelial cells into this cancer cell-free extracellular matrix microenvironment (Fig. 3e and Supplementary Fig. 6).

4) *In Fig. 4, the authors nicely show the enhanced survival effect of combined targeting of VEGF (with an antibody to VEGF) and an inhibitor of EGFR in the xenograft model of GBM, and the authors also show that this combined targeting appears to block formation of the large diameter blood vessels in the tumors. The authors need to show decreased phosphorylation of EGFR/EGFRvIII in these tumor vessels to convincingly demonstrate specific inhibition of pEGFR with the EGFR inhibitor.*

This is an excellent suggestion. While this approach is technically challenging due to the EGFR signal of adjacent cancer cells that overwhelmed that of endothelial cells in our imaging attempts, we were eventually able to stain the vessels of MES-GSC xenografts using rabbit anti-pEGFR Tyr992 specific antibody recognizing the tyrosine kinase domain of EGFR in the presence or absence of Dacomitinib (Supplementary Fig. 17b).

It should also be mentioned that at the concentrations we used, Dacomitinib is very specific for ErbB family of kinases including EGFR³¹. Also, in our hands, the effects of this agent were virtually indistinguishable from *EGFR* gene targeting, which we hope further strengthens the notion that the results we obtained are EGFR-specific.

5) *The spatial transcriptomic findings from the 8 xenograft tumors (Fig. 5) where regions of interest (ROIs) showed differences in gene expression needs some validation of the key genes discussed in the manuscript as being relevant to the vascular processes focused on.*

This is an important point which we approached through several experiments now included in the revised version of our manuscript and in this letter. While we obtained largely encouraging results using spatial sequencing, upon extensive validation the GeoMX platform was found to lack single-cell resolution and we revised our interpretation of previously reported results. Thus, GeoMX was excellent at separating EGFR-positive from EGFR-negative xenografts, but some of the expected signals originally ascribed to blood vessels were ultimately found in cells other than endothelium (stroma cells). We believe that these data have value as descriptive illustration of regional and perivascular responses to the EGFR status but were insufficient to unequivocally identify hallmarks specific for endothelial cells associated with *vasectasia*. We revised the text accordingly. The results

obtained with GeoMX platform prompted us to rely on single cell RNA sequencing to define the signature of endothelial cells involved in angiogenesis and *vasectasia*.

Nonetheless, based on clues from both GeoMX and scRNAseq analysis, we decided to perform two levels of validations: (i) immunostaining for protein products of selected top unique genes differentially expressed between EGFR-WT vs EGFR-KO tumours and enriched in CD31 positive endothelial cell, mainly from single cell sequencing data; (ii) We performed RNAscope mRNA profiling focusing on top targets and carried out on the corresponding xenograft tissues.

The results of immunostaining were especially striking, as we found that angiogenic endothelium in EGFR-KO tumours was enriched, as expected, for VEGFR2 and Apelin proteins, which are markers of angiogenic tip cells. At the same time the *vasectasia*-related blood vessels in EGFR-WT xenografts were negative for these tip cell markers, but instead expressed ample amounts of Survivin/Birc5, Socs2 and Srsf2, which were enriched in corresponding scRNAseq dataset (Fig. 4g and Supplementary Fig. 15a).

The RNAscope Fluorescent Multiplex In-Situ hybridisation assays were also carried out in the corresponding MES-GSC xenografts in mice, and they revealed endothelial expression of several candidate transcripts implicated in *vasectasia*, such as Survivin/Birc5, Socs2 and Srsf2 (Supplementary Fig. 15b).

We hope these new results and revised interpretation will be found convincing. We are most grateful that Reviewer stimulated us to perform this validation and shared comments that helped us improve the paper in many other ways.

Reviewer #4:

The report by C. Spinelli et al elegantly demonstrated EGFR-dependent neovascularization of MES or PN-GBM subtypes. The main novelty of their research is the role of EVs containing EGFR mRNA, causing expression of tumor-associated EGFR on endothelial cells and subsequently pathological neovascularization. Based on the transcriptional subtype and potential occurrence of Chr 7 and EGFR amplifications, the tumors acquired distinct vascularization patterns, ranging from large vessels, namely vasectasia (EGFR-high), to very dense vascular networks (EGFR-low). The paper is well written, and its hypotheses are clearly stated. The experiments are designed in a comprehensible way. In my opinion, the overall work is conclusive and addresses an important topic for the scientific community.

We would like to thank the Reviewer for this incisive synopsis of our work and a supportive and favourable assessment of its qualities. We have certainly striven to document our novel findings, as thoroughly as we could.

Despite the already mature impression of the work, I have some concerns and weaknesses that ought to be addressed in a revision.

1. *the model and the transfer to humans. The main problem is the exclusive use of murine models (despite the use of human cell cultures). The authors might want to use some human data for validation or explore their hypothesis and results in a human setting. The use of cell type-specific transcriptomics is interesting and can be easily extended to human FFPE samples. What is the effect of BEV therapy in human samples? Or at least use some human datasets, such as the latest single cell or spatial transcriptome data (for stRNA-seq, H&E could also help identify the different types/patterns of neovascularization).*

We completely understand and share Reviewer's thoughts, as to the translation (interpretation) of our results in the context of human disease. We have invested considerable efforts in addressing this question, as best we could, and we highlighted the limitations of our approaches in the revised version of the manuscript. Perhaps one point to be made, however, before we delve into the related specifics, would be to say that while questions of human relevance are inevitably associated with any animal and cellular model of cancer, these models, due to their simplicity, often reveal processes that are otherwise obscured (and missed) in a more complex human tissue microenvironment.

To present solid evidence, as to the existence, role and contributions of EGFR-EV-driven *vasectasia* in human GBM, is challenging and will probably require more than one study. In the clinical reality *vasectasia* likely co-exists with angiogenesis and other heterogeneous vascular processes (cooption, microthrombosis, vascular regression³²), as do multiple subtypes of their driving cancer cells and GSCs^{33,34}. Therefore, to some extent, these different mechanisms need to be parsed out (as we did through modelling) to be dissected and understood in more detail.

Nonetheless, our model predicts that a subset of endothelial cells in GBM should express EGFR. We posit also that targeting the canonical, VEGF-driven angiogenesis pathway should result in selection against small angiogenic blood vessels and the enrichment of enlarged (*vasectatic*) vessels. To this end, we have been able to document that, indeed, in human GBM single cell datasets a subset of CD34-positive endothelial cells expresses EGFR mRNA (Supplementary Fig. 13c).

Addressing the question whether in human GBM, targeting VEGF would lead to diminution of the angiogenic microvasculature and to a preponderance of alternative processes such as *vasectasia*, we found to be extremely challenging for several reasons. First, anti-VEGF therapy is currently only used in recurrent GBM settings, and those lesions are infrequently surgically resected a situation resulting in the respective tissue samples and matching controls (e.g. first resection of the same tumour) being virtually unavailable for our studies. In spite of multiple attempts to engage collaborating neuropathologists in several large centres, we received no analyzable material to study vascular patterns in such samples. Again, this would have been mostly recurrent lesions.

Second, by their very nature, the recurrent lesions would represent a fundamentally different scenario than that our study focuses on. This is because recurrent GBM would arise from cells (GSCs) that have sustained multiple genotoxic insults due to prior temozolomide chemotherapy and radiation, currently used as a standard of care³⁵. This

natural history is known to result in the extensive damage to the cellular genome ³⁶ and major changes in cell phenotype, production of EVs ³⁷ and likely also in the repertoire of the angiogenic secretome. This is in contrast to our experiments, in which glioma stem cells are therapy naïve and reveal their inherent and unperturbed potentials to interact with the vasculature in a subtype specific manner.

Third point we would like to suggest is that Bevacizumab has been used in settings of treatment naïve GBM, as a part of phase III clinical trials, such as AvaGlio and RTOG0825 ^{17,38}, albeit in the context of temozolomide and radiation. Regardless of the likely interference of standard therapies, the related tissue material is currently not accessible to us for direct analysis. However, similar GBM tissues were recently analysed by Blumenthal et al in a relatively unusual cohort of patients who received Bevacizumab and were subsequently operated at the time of recurrence ¹⁸. Interestingly, while the authors do not comment on this directly, a careful review of this study indicates that following anti-VEGF therapy tumours were predictably depleted for capillary networks, but histological images reveal the presence of mostly larger vessels resembling *vasectasia* that we have observed in mice (*Fig. 1* in Blumenthal et al. 2018).

This may suggest (but does not prove) that large, non-angiogenic vessel formation may enable tumour escape from Bevacizumab therapy. However, this notion remains to be documented in terms of whether the post-Bevacizumab vascular pattern represents cooption of static pre-existing vessels, or formation of actively growing enlarged vessels characteristic of *vasectasia*. Blumenthal et al do not comment on the nature of these vascular patterns, or on the expression of oncogenic EGFR in recurrent lesions being analysed.

With the knowledge obtained through our present study, including new data generated during this revision, it may now be possible to explore functional and molecular interactions and hallmarks of *vasectasia*, such as the expression of EGFR, Birc5 and Ki67 in large vessels of post-Bevacizumab human GBM samples, should they become available to us.

Still, we believe (and this is highlighted here, and in the text) that our results offer a novel way to look at GBM vasculature, beyond angiogenesis, and to seek non-canonical forms of neovascularization and their functional consequences. Arguably, the profound molecular differences in endothelial subpopulations (phenotypes) that emerge under the influence of EGFR-EVs during *vasectasia* (vs angiogenesis) likely impact processes beyond blood perfusion and ‘feeding’ tumour tissue. These unique and cancer-specific cellular states may impact such fundamental endothelial functions as immunoregulation, drug penetration, therapeutic responses or paracrine (angiocrine) interactions with tumour parenchyma ³⁹. We are pursuing some of these themes for future extensions of our present study, and we hope the Reviewer would kindly consider the possible implications of this work even if we cannot conclusively and directly determine the impact of *vasectasia* on clinical GBM at this stage.

2. *cellular and microenvironmental heterogeneity.* The well-known cellular heterogeneity of GBMs is almost ignored in the presented work. I fully understand the use of pre-classified MES/PN cell lines representing the majority of neural/oligo- or mesenchymal differentiated cell states for a model, but this limitation needs to be discussed in more detail. The different pattern of vascularization can also exist in the same tumor, suggesting that spatial heterogeneity or other spatially resolved confounding factors (such as metabolism or immunity) may influence or interact with the described results.

This is an extremely important and valuable point. It was never our intention to ignore or underplay the heterogeneity of GBMs or the role of the tumour microenvironment. In fact we are extremely interested in the emerging models of this class of diseases, where single cell RNA sequencing and other methods revealed a complex hierarchical mosaic of phenotypic states with oncogene-induced biases governing the numerical equilibria that traditionally underly GBM subtypes^{33,40}. We apologise for being laconic about this in the original submission and have expanded on this question in our revisions as much as possible.

We also completely agree with the Reviewer that our reliance of pre-classified GSC xenografts limits our ability to recapitulate the whole picture of GBM in any one of these models, whether proneural or mesenchymal. While this is a limitation in a ‘synthetic’ sense, we would argue, it is also an ‘analytical’ opportunity. This is because MES-GSC or PN-GSC isolates (‘generators’ of EGFR-EV and VEGF signals, respectively) enable us to deconstruct individual elements within the GBM cellular mosaic and understand their individual ways of stimulating the vasculature, i.e. mainly by *vasectasia* or angiogenesis, respectively. This, we believe, is a missing stepping stone to be able to deconvolute the spatial complexity of vascular responses in actual GBMs. At the very least, the lesson from such an analysis is that more than one vascular process needs to be considered and possibly targeted or modulated for therapeutic purposes. We would also suggest that non-angiogenic vascular patterns, such as *vasectasia*, result from active growth process and that large vessels in GBM may not simply represent engulfment of static pre-existing pre-capillaries, but an active response to ectopic stimulation with tumour cell derived EVs.

We absolutely concur with the Reviewer that there could be confounders that may obscure the effects of EV-EGFR (or other factors) on the global patterns of the vasculature in GBM, including metabolic pressures, hypoxia, infiltration with immune cells, therapeutic insults and other events. We added this important consideration to our discussion, as much as possible. Still, we believe that knowing that processes such as *vasectasia* may exist and, as we reported, may be identified by some of their markers (Birc5, EGFR, morphology) and mechanisms (EGFR-EVs), could help better understand GBM microenvironment and possibly design additional interventions. For example, as mentioned in the previous paragraph the multiple regulatory and barrier effects (e.g. for immune cells) of the endothelium in GBM would likely be altered by phenotypic change induced by *vasectasia*. We do not yet have data to document this aspect, but we hope to convince the Reviewer that our present work enables us and others to move in that direction.

3. *Genetic background and expression.* EGFR, together with Chr7 amplification, is the

hallmark alteration of GBM. Cell lines often lose their initial high EGFR copy gain to a more moderate level. What is the level of EGFR gain in the cell lines studied? Is there a correlation between genetic (CNA) and EGFR (mRNA+) EVs? It would be interesting to study the vascular pattern of tumors with higher or lower EGFR Amp.

This is an excellent question, which would be of great interest to address. We previously performed DNA sequencing of some of our GSC lines ⁴¹ and re-examination of this dataset revealed multiple copies of EGFR gene, along with abundant transcript and protein ³⁷. Since these cells were maintained in a sphere culture and in serum free media, their phenotypes remained remarkably stable, including EGFR expression, and amplification (data not shown). This observation is consistent with the current literature ⁴², as improved culturing techniques have remedied somewhat the loss of EGFR amplification, that indeed, was a source of experimental artifacts in the past ⁴³. It would be, of course, of interest to assess whether GSCs (especially MES-GSCs) exhibit fewer copies of EGFR (and chr.7 of extrachromosomal DNA) than original tumours, but we do not presently have access to this material that was originally processed by Dr. Ichiro Nakano who initially described ³⁰, and kindly supplied us with these GSC lines.

We also agree that correlating vascular patterns in human GBM series with chr7 amplification and/or expression of EGFR and/or EGFRvIII mutation would be a fascinating subject for future studies, especially with access to a larger, properly powered cohort of tissue samples. Since our results suggest that selective restoration of EGFRvIII into the EGFR-knock out cells (GSC83, GSC1005) is sufficient to restore *vasectasia* in xenografts we would predict that the level of this transcript may be crucial for this process to occur. However, at the present time, we do not have access to experimental tools to study in controlled manner whether the level of chromosome 7 amplification or extrachromosomal EGFR sequences quantitatively correlate with the ability to trigger *vasectasia* and what are the thresholds. This would be a fascinating project for future studies, and we thank the Reviewer for bringing this to our attention.

In conclusion, the work is of great interest. The translation to human tumors and various confounding factors based on tumor heterogeneity or microenvironment should be further explored and discussed.

We would like to thank the Reviewer for outstanding feedback and excellent suggestions, which we have implemented, as much as we could, given available resources, time and extent of data already included in the manuscript as well as space available for discussion. We believe that our study, the first description of EGFR-EV-driven *vasectasia*, is a crucial stepping stone for our future analyses of complex human GBM tissues (and possibly vascular events in other cancers) to more fully understand and harness vascular responses and their oncogenic triggers, beyond angiogenesis. Naturally, the modulating influences of the dynamic tumour microenvironment presents a larger context for these considerations.

In closing, we are immensely grateful for the input of all four Reviewers, who provided us with constructive, courteous, and thoughtful feedback, which led to much additional work and considerable improvement of our submission. We would also like to thank the Editors

for the opportunity to re-submit our work for re-evaluation. Finally, we hope that our revised manuscript is now stronger, and would be found acceptable for publication in Nature Communications.

References used:

- 1 Nicosia, R. F. The aortic ring model of angiogenesis: a quarter century of search and discovery. *J Cell Mol Med* **13**, 4113-4136 (2009). <https://doi.org/10.1111/j.1582-4934.2009.00891.x>
- 2 Gerhardt, H. *et al.* VEGF guides angiogenic sprouting utilizing endothelial tip cell filopodia. *J Cell Biol* **161**, 1163-1177 (2003). <https://doi.org/10.1083/jcb.200302047>
- 3 DeCicco-Skinner, K. L. *et al.* Endothelial cell tube formation assay for the in vitro study of angiogenesis. *J Vis Exp*, e51312 (2014). <https://doi.org/10.3791/51312>
- 4 Betsholtz, C. Cell-cell signaling in blood vessel development and function. *EMBO Mol Med* **10** (2018). <https://doi.org/10.15252/emmm.201708610>
- 5 Spinelli, C. *et al.* Molecular subtypes and differentiation programmes of glioma stem cells as determinants of extracellular vesicle profiles and endothelial cell-stimulating activities. *J Extracell Vesicles* **7**, 1490144 (2018). <https://doi.org/10.1080/20013078.2018.1490144>
- 6 Aissa, A. F. *et al.* Single-cell transcriptional changes associated with drug tolerance and response to combination therapies in cancer. *Nat Commun* **12**, 1628 (2021). <https://doi.org/10.1038/s41467-021-21884-z>
- 7 Carmeliet, P. & Jain, R. K. Molecular mechanisms and clinical applications of angiogenesis. *Nature* **473**, 298-307 (2011).
- 8 Sathornsumetee, S. *et al.* Phase II trial of bevacizumab and erlotinib in patients with recurrent malignant glioma. *Neuro Oncol* **12**, 1300-1310 (2010). <https://doi.org/10.1093/neuonc/noq099>
- 9 Hegi, M. E. *et al.* Pathway analysis of glioblastoma tissue after preoperative treatment with the EGFR tyrosine kinase inhibitor gefitinib—a phase II trial. *Mol Cancer Ther* **10**, 1102-1112 (2011). <https://doi.org/10.1158/1535-7163.MCT-11-0048>
- 10 Sepulveda-Sanchez, J. M. *et al.* Phase II trial of dacomitinib, a pan-human EGFR tyrosine kinase inhibitor, in recurrent glioblastoma patients with EGFR amplification. *Neuro-oncology* **19**, 1522-1531 (2017). <https://doi.org/10.1093/neuonc/nox105>
- 11 Chi, A. S. *et al.* Exploring Predictors of Response to Dacomitinib in EGFR-Amplified Recurrent Glioblastoma. *JCO Precis Oncol* **4** (2020). <https://doi.org/10.1200/po.19.00295>
- 12 Takahashi, T. *et al.* Phase I and pharmacokinetic study of dacomitinib (PF-00299804), an oral irreversible, small molecule inhibitor of human epidermal growth factor receptor-1, -2, and -4 tyrosine kinases, in Japanese patients with advanced solid tumors. *Invest New Drugs* **30**, 2352-2363 (2012). <https://doi.org/10.1007/s10637-011-9789-z>
- 13 Creamer, B. A. *et al.* Associations between Pregnane X Receptor and Breast Cancer Growth and Progression. *Cells* **9** (2020). <https://doi.org/10.3390/cells9102295>
- 14 Jain, R. K. *et al.* Angiogenesis in brain tumours. *Nat Rev Neurosci* **8**, 610-622 (2007). <https://doi.org/10.1038/nrn2175>
- 15 Reardon, D. A. *et al.* A review of VEGF/VEGFR-targeted therapeutics for recurrent glioblastoma. *J Natl Compr Canc Netw* **9**, 414-427 (2011). <https://doi.org/10.6004/jnccn.2011.0038>
- 16 Gilbert, M. R. *et al.* A randomized trial of bevacizumab for newly diagnosed glioblastoma. *N Engl J Med* **370**, 699-708 (2014). <https://doi.org/10.1056/NEJMoa1308573>
- 17 Chinot, O. L. *et al.* AVAglio: Phase 3 trial of bevacizumab plus temozolomide and radiotherapy in newly diagnosed glioblastoma multiforme. *Adv Ther* **28**, 334-340 (2011). <https://doi.org/10.1007/s12325-011-0007-3>

- 18 Blumenthal, D. T. *et al.* Surgery for Recurrent High-Grade Glioma After Treatment with Bevacizumab. *World Neurosurg* **110**, e727-e737 (2018). <https://doi.org/10.1016/j.wneu.2017.11.105>
- 19 Al-Nedawi, K., Meehan, B., Kerbel, R. S., Allison, A. C. & Rak, J. Endothelial expression of autocrine VEGF upon the uptake of tumor-derived microvesicles containing oncogenic EGFR. *Proc Natl Acad Sci U S A* **106**, 3794-3799 (2009). <https://doi.org/10.1073/pnas.0804543106>
- 20 Al-Nedawi, K. *et al.* Intercellular transfer of the oncogenic receptor EGFRvIII by microvesicles derived from tumour cells. *Nat Cell Biol* **10**, 619-624 (2008). <https://doi.org/10.1038/ncb1725>
- 21 Tran, J. *et al.* A role for survivin in chemoresistance of endothelial cells mediated by VEGF. *Proc Natl Acad Sci U S A* **99**, 4349-4354 (2002). <https://doi.org/10.1073/pnas.072586399>
- 22 Altieri, D. C. Survivin, cancer networks and pathway-directed drug discovery. *Nat Rev Cancer* **8**, 61-70 (2008). <https://doi.org/10.1038/nrc2293>
- 23 Asanuma, H. *et al.* Survivin expression is regulated by coexpression of human epidermal growth factor receptor 2 and epidermal growth factor receptor via phosphatidylinositol 3-kinase/AKT signaling pathway in breast cancer cells. *Cancer Res* **65**, 11018-11025 (2005). <https://doi.org/10.1158/0008-5472.Can-05-0491>
- 24 Keating, N. & Nicholson, S. E. SOCS-mediated immunomodulation of natural killer cells. *Cytokine* **118**, 64-70 (2019). <https://doi.org/10.1016/j.cyto.2018.03.033>
- 25 Goldshmit, Y., Walters, C. E., Scott, H. J., Greenhalgh, C. J. & Turnley, A. M. SOCS2 induces neurite outgrowth by regulation of epidermal growth factor receptor activation. *J Biol Chem* **279**, 16349-16355 (2004). <https://doi.org/10.1074/jbc.M312873200>
- 26 Wan, J., Che, Y., Kang, N. & Wu, W. SOCS3 blocks HIF-1 α expression to inhibit proliferation and angiogenesis of human small cell lung cancer by downregulating activation of Akt, but not STAT3. *Mol Med Rep* **12**, 83-92 (2015). <https://doi.org/10.3892/mmr.2015.3368>
- 27 Li, K. & Wang, Z. Splicing factor SRSF2-centric gene regulation. *Int J Biol Sci* **17**, 1708-1715 (2021). <https://doi.org/10.7150/ijbs.58888>
- 28 Abou Faycal, C., Gazzeri, S. & Eymine, B. A VEGF-A/SOX2/SRSF2 network controls VEGFR1 pre-mRNA alternative splicing in lung carcinoma cells. *Sci Rep* **9**, 336 (2019). <https://doi.org/10.1038/s41598-018-36728-y>
- 29 Yadav, P. *et al.* Hypoxia-induced loss of SRSF2-dependent DNA methylation promotes CTCF-mediated alternative splicing of VEGFA in breast cancer. *iScience* **26**, 106804 (2023). <https://doi.org/10.1016/j.isci.2023.106804>
- 30 Mao, P. *et al.* Mesenchymal glioma stem cells are maintained by activated glycolytic metabolism involving aldehyde dehydrogenase 1A3. *Proc Natl Acad Sci U S A* **110**, 8644-8649 (2013). <https://doi.org/10.1073/pnas.1221478110>
- 31 Engelman, J. A. *et al.* PF00299804, an irreversible pan-ERBB inhibitor, is effective in lung cancer models with EGFR and ERBB2 mutations that are resistant to gefitinib. *Cancer Res* **67**, 11924-11932 (2007).
- 32 Holash, J., Wiegand, S. J. & Yancopoulos, G. D. New model of tumor angiogenesis: dynamic balance between vessel regression and growth mediated by angiopoietins and VEGF. *Oncogene* **18**, 5356-5362 (1999). <https://doi.org/10.1038/sj.onc.1203035>
- 33 Patel, A. P. *et al.* Single-cell RNA-seq highlights intratumoral heterogeneity in primary glioblastoma. *Science* **344**, 1396-1401 (2014). <https://doi.org/10.1126/science.1254257>
- 34 Nakano, I. Proneural-mesenchymal transformation of glioma stem cells: do therapies cause evolution of target in glioblastoma? *Future Oncol* **10**, 1527-1530 (2014). <https://doi.org/10.2217/fon.14.86>
- 35 Wen, P. Y. *et al.* Glioblastoma in adults: a Society for Neuro-Oncology (SNO) and European Society of Neuro-Oncology (EANO) consensus review on current management

- and future directions. *Neuro Oncol* **22**, 1073-1113 (2020).
<https://doi.org:10.1093/neuonc/noaa106>
- 36 Wang, J. *et al.* Clonal evolution of glioblastoma under therapy. *Nat Genet* **48**, 768-776
(2016). <https://doi.org:10.1038/ng.3590>
- 37 Garnier, D. *et al.* Divergent evolution of temozolomide resistance in glioblastoma stem
cells is reflected in extracellular vesicles and coupled with radiosensitization. *Neuro Oncol*
20, 236-248 (2018). <https://doi.org:10.1093/neuonc/nox142>
- 38 Weller, M. & Yung, W. K. Angiogenesis inhibition for glioblastoma at the edge: beyond
AVAGlio and RTOG 0825. *Neuro Oncol* **15**, 971 (2013).
<https://doi.org:10.1093/neuonc/not106>
- 39 Adnani, L. *et al.* Angiocrine extracellular vesicles impose mesenchymal reprogramming
upon proneural glioma stem cells. *Nat Commun* **13**, 5494 (2022).
<https://doi.org:10.1038/s41467-022-33235-7>
- 40 Neftel, C. *et al.* An Integrative Model of Cellular States, Plasticity, and Genetics for
Glioblastoma. *Cell* **178**, 835-849 e821 (2019). <https://doi.org:10.1016/j.cell.2019.06.024>
- 41 Daniel, P. *et al.* Detection of temozolomide-induced hypermutation and response to PD-1
checkpoint inhibitor in recurrent glioblastoma. *Neurooncol Adv* **4**, vdac076 (2022).
<https://doi.org:10.1093/noajnl/vdac076>
- 42 Furnari, F. B., Cloughesy, T. F., Cavenee, W. K. & Mischel, P. S. Heterogeneity of
epidermal growth factor receptor signalling networks in glioblastoma. *Nat Rev Cancer* **15**,
302-310 (2015). <https://doi.org:10.1038/nrc3918>
- 43 Bigner, S. H. *et al.* Characterization of the epidermal growth factor receptor in human
glioma cell lines and xenografts. *Cancer Res* **50**, 8017-8022 (1990).

EVIEWERS' COMMENTS

Reviewer #2 (Remarks to the Author):

The authors have successfully addressed the majority of the comments raised during the initial review, resulting in substantial improvements to the manuscript. While I note the validation using human specimens from external resources remains somewhat limited, the overall quality and contributions of the work are significant. Therefore, I recommend acceptance of the manuscript for publication in this journal.

Reviewer #3 (Remarks to the Author):

This revised manuscript beautifully describes a new vascular process in endothelial cells that appears to be driven by EGFR/EGFRvIII found in the extracellular vesicles (EVs) of mesenchymal glioma stem cells. The authors have satisfactorily addressed my concerns and comments with both the addition of new data and with clarification. The authors now clarify in the manuscript that in a general sense the neovascularization process is more complex than we previously thought, and the type of neovascularization process described here termed “vasectasia” is distinct from the well-described and classical neovascularization process driven by VEGF-A. These findings definitely have relevance to disease and to anti-angiogenic therapy. The revised manuscript contains highly novel findings that are relevant to our understanding of the complexity of the vascularization process in cancer, and the manuscript should be accepted.

Minor Concerns:

- 1) In Figure 5b, survival plot of tumor-bearing mice: The colors for the lines representing the various mouse group conditions are too close to each other. I suggest using more distinct colors for the survival lines of the mouse groups.
- 2) In Figure 5c, images of immunofluorescence of the mouse tumors from Figure 5B: There need to be lines between the individual images to help a reader distinguish what blood vessels belong to what mouse treatment group.

Reviewer #4 (Remarks to the Author):

No further comments.

Point-by-point response to reviewers' comments:

We would like to express our utmost gratitude to all Reviewers for their thoughtfulness, patience, guidance and constructive critique that helped us to considerably improve our manuscript. All this is greatly appreciated.

Reviewer #2 (Remarks to the Author):

The authors have successfully addressed the majority of the comments raised during the initial review, resulting in substantial improvements to the manuscript. While I note the validation using human specimens from external resources remains somewhat limited, the overall quality and contributions of the work are significant. Therefore, I recommend acceptance of the manuscript for publication in this journal.

We are grateful for Reviewer's supportive comments. We agree that the process of validating our findings in human tissues has only just begun. This effort will be facilitated by the results we have obtained thus far, especially molecular characteristics of vasectasia and a better understanding of this fascinating process. We believe that the current biological analysis represents a necessary step in further, more translational, explorations, and we are grateful that Reviewer sees this in a similar light. A thorough clinical validation (contextualization) of our findings will amount to an extensive independent undertaking (as it often is for new biological processes). The completion of this effort will depend on accessing new and well annotated biobanks of tissue material, generating additional spatial information and formation of additional collaborations, all of which we are in the process of putting in place. We appreciate enabling us to prepare for these steps.

Reviewer #3 (Remarks to the Author):

This revised manuscript beautifully describes a new vascular process in endothelial cells that appears to be driven by EGFR/EGFRvIII found in the extracellular vesicles (EVs) of mesenchymal glioma stem cells. The authors have satisfactorily addressed my concerns and comments with both the addition of new data and with clarification. The authors now clarify in the manuscript that in a general sense the neovascularization process is more complex than we previously thought, and the type of neovascularization process described here termed "vasectasia" is distinct from the well-described and classical neovascularization process driven by VEGF-A. These findings definitely have relevance to disease and to anti-angiogenic therapy. The revised manuscript contains highly novel findings that are relevant to our understanding of the complexity of the vascularization process in cancer, and the manuscript should be accepted.

We would like to thank the Reviewer for this incisive and positive assessment of our work. We are, naturally, quite excited about our novel observations and hope that they would extend the

biological understanding of vascular processes in cancer (especially GBM), while also opening some new translational opportunities, precisely along the lines articulated in Reviewer's comments. Of course, much remains to be studied, but we believe that we provided a solid foundation to refine the possible translational hypotheses and delve into previously overlooked complexities of tumour-vascular interactions. Again, we thank the Reviewer for kind words and recognition of our advances.

Minor Concerns:

1) In Figure 5b, survival plot of tumor-bearing mice: The colors for the lines representing the various mouse group conditions are too close to each other. I suggest using more distinct colors for the survival lines of the mouse groups.

Yes, this is a valid point, and the colours and line styles in Figure 5b have been adjusted to make them more distinguishable. We hope the effects are satisfactory.

2) In Figure 5c, images of immunofluorescence of the mouse tumors from Figure 5B: There need to be lines between the individual images to help a reader distinguish what blood vessels belong to what mouse treatment group.

Absolutely, we agree with this concern, and have modified the images to ensure that individual fields are well separated and clearly depict the intended details. We are grateful for these helpful comments and a positive reception of our work.

Reviewer #4 (Remarks to the Author):

No further comments.

We are grateful for this statement that we interpret as expressing satisfaction with our efforts to improve our paper. We very much appreciate Reviewer's intellectual guidance through this study and helping us to bring it to a solid conclusion.